# Denoising Neural Reranker for Recommender Systems

**Wenyu Mao**[1][*] **Shuchang Liu**[2][‡] **Hailan Yang**[2]**, Xiaobei Wang**[2]**, Xiaoyu Yang**[2]**, Xu Gao**[2]**,
Xiang Li**[2]**, Lantao Hu**[2]**, Han Li**[2]**, Kun Gai**[2]**, An Zhang**[1]**, Xiang Wang**[1][†]
[1] University of Science and Technology of China
[2] Kuaishou Technology

## Abstract

For multi-stage recommenders in industry, a user request would first trigger a simple and efficient retriever module that selects and ranks a list of relevant items, then the recommender calls a slower but more sophisticated reranking model that refines the item list exposure to the user. To consistently optimize the two-stage retrieval reranking framework, most efforts have focused on learning reranker-aware retrievers. In contrast, there has been limited work on how to achieve a retriever-aware reranker. In this work, we provide evidence that the retriever scores from the previous stage are informative signals that have been underexplored. Specifically, we first empirically show that the reranking task under the two-stage framework is naturally a noise reduction problem on the retriever scores, and theoretically show the limitations of naive utilization techniques of the retriever scores. Following this notion, we derive an adversarial framework DNR that associates the denoising reranker with a carefully designed noise generation module. The resulting DNR solution extends the conventional score error minimization loss with three augmented objectives, including: 1) a denoising objective that aims to denoise the noisy retriever scores to align with the user feedback; 2) an adversarial retriever score generation objective that improves the exploration in the retriever score space; and 3) a distribution regularization term that aims to align the distribution of generated noisy retriever scores with the real ones. We conduct extensive experiments on three public datasets and an industrial recommender system, together with analytical support, to validate the effectiveness of the proposed DNR. The code is released at `https://github.com/maowenyu-11/DNR`.

## 1 Introduction

Recommender systems aim to generate personalized item lists and maximize users' engagement. For large-scale item pools, industrial solutions Ricci et al. (2022); Liu et al. (2025) typically use a two-stage retriever-reranker architecture. The retriever efficiently narrows down a large item pool to a manageable relevant candidate set. The reranker then uses a more sophisticated model to reorder the candidates into an optimal list-wise order. To optimize the two-stage framework towards a consistent goal of recommendation (*e.g.,* aligning with user behavior or preference) across all stages, the majority of efforts have been focusing on learning a reranker-aware retriever Gallagher et al. (2019); Xu et al. (2024a). However, there is limited work on how to achieve a retriever-aware reranker.

For the latter reranker Carbonell & Goldstein (1998); Jin et al. (2008); Pei et al. (2019); Liu et al. (2022), early studies propose to re-score the retrieval items Jin et al. (2008); Ai et al. (2018); Pei et al. (2019) by modeling the mutual influences among candidate items. Recent state-of-the-art approaches find it more effective to formulate the reranking task as a list generation problem Shi et al. (2023); Ren et al. (2024); Lin et al. (2024), where the lists with better list-wise rewards are given higher generation probability. Yet, existing methods largely overlook the potential of initial retriever scores, which may offer rich prior information for the reranker stage. In practice, a straightforward approach

---

[*]Work done during the internship at Kuaishou: wenyumao2@gmail.com
[†]Corresponding author: liushuchang@kuaishou.com, xiangwang@ustc.edu.cn
[‡]The first two authors contributed equally to this work

to leverage initial retriever scores is to directly include them as extra input features of the model, which shows promise to improve reranking performance as in Figure 1a. Still, as we will illustrate in section 3.2, this naive solution might be far from exploiting this retriever information effectively to align with user feedback, which is addressed by the following contributions in this work.

**Formulating reranking as a noise reduction problem:** The retriever stage typically employs a simpler model than the reranker to process large candidate pools, given computational budget constraints. In contrast, the reranker, operating on a much smaller candidate set, can use a more sophisticated architecture and thus achieves significantly higher accuracy (Figure 1a, 1b). What follows is an empirical assumption that there is higher noise in the retriever scores than the reranker scores, which naturally indicates a *noise reduction process* of the reranking stage. This noise reduction pattern is also evident when we observe the changes in the embedding representations of ground truth target items. We show evidence by comparing Figure 1c and Figure 1d, where the same set of retrieved candidates is not distinguished well by the retriever, but the target items are better predicted by the reranker, in terms of the embedding space. The reranker's noise is more concentrated near zero than the retriever's (Figure 1e), further confirming the reranker acts as a denoising model for the retriever's noise. However, due to the uncertainty inherent in noisy retriever scores, denoising based on these scores may misalign with the system-wide goal of aligning with user behavior. Thus, we argue that learning a noise-aware reranker is crucial for achieving a retriever-aware reranker.

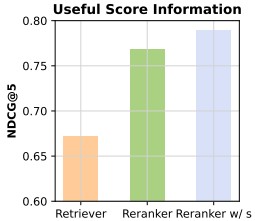

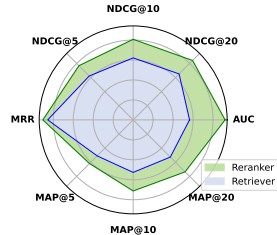

(a) Initial retriever scores further boost the performance.

(b) Performance comparison of two stages in MovieLens dataset.

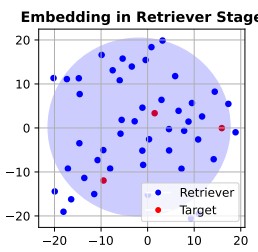

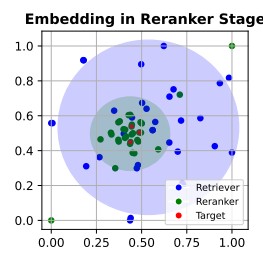

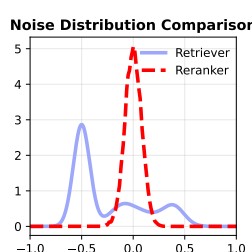

(c) Target item and selected candidates embedding in retriever.

(d) Noise reduction process in item embedding perspective.

(e) Noise density distribution of the reranker and retriever.

Figure 1: (a) The informative retriever score and (b-d) noise reduction nature of reranking. The reranker in this example uses transformer model to select the top-20 items from the candidates. "Rerank w/ s" represents using retriever scores as additional item features for the reranker. The black shaded circles in (c) and (d) represent the retriever's selection of candidate items, and the green shaded circle in (d) represents those of the reranker for exposure. (e) compares the noise distribution (distance between predicted scores and ground-truth labels) of the retriever and the reranker.

**The proposed solution:** Motivated by the previous analysis, we propose an adversarial framework, which formulates a *denoising neural reranker (DNR)* with a carefully designed *noise generation module*, which introduces unseen noise to users' feedback to augment the distribution modeling of retriever scores. We theoretically show that the user behavior alignment goal can be decomposed into three learning objectives in our solution framework: 1) a denoising objective that aims to denoise the noisy retriever scores under both observed and synthetic retriever scores; 2) an adversarial objective that encourages the noise generator to synthesize adversarial samples that are hard to denoise, which in turn improves the reranker's effectiveness; and 3) a score distribution regularization term that aims to align the distribution between synthetic retriever scores with the real retriever scores.

We conduct empirical experiments on three public datasets and an industrial recommender system to verify the superiority of DNR against naive utilization strategies of retriever scores, as well as leading

reranker models. Additionally, we find that the three proposed learning objectives in the adversarial denoising framework yield better recommendation performance than a single denoising objective.

## 2 RELATED WORK

### 2.1 MULTI-STAGE RECOMMENDATION

Modern content-intensive web services with large candidate pools typically employ multi-stage recommender systems Qin et al. (2022); Zheng et al. (2024); Liu et al. (2022) (also known as cascade ranking Liu et al. (2017); Gallagher et al. (2019) process) to balance accuracy and efficiency under strict latency constraints. Each stage progressively narrows down the candidate set while employing increasingly sophisticated models. While most existing work focuses on independently optimizing individual stages (*e.g.,* retrieval Wang et al. (2011; 2017) or the reranking stage Liu et al. (2022); Pei et al. (2019)), recent studies explore joint optimization across the entire pipeline Gallagher et al. (2019); Qin et al. (2022); Xu et al. (2024b). These approaches leverage later-stage models to guide earlier-stage learning, improving overall coherence and performance Zhao et al. (2023); Xu et al. (2024a). In contrast, our work takes a complementary direction——using earlier-stage scores to regulate later-stage optimization.

### 2.2 RERANKING STRATEGIES

In sequential recommendation, re-ranking Pei et al. (2019); Huzhang et al. (2023); Ai et al. (2018) aims to refine the initial ranking list generated by a previous retrieval stage. Existing re-ranking approaches can be broadly categorized into the following paradigms. Single-point re-scoring methods Kang & McAuley (2018); Zhuang et al. (2018) independently predict a refined score for each item and re-rank them accordingly. While efficient, they ignore dependencies between items, potentially leading to suboptimal list-level performance. Unlike single-point approaches, list-refinement methods Pei et al. (2019); Ai et al. (2018); Xi et al. (2022) explicitly model mutual influences between items by treating the pre-ranked candidate list as input. Techniques such as Transformer-based architectures Pei et al. (2019) capture user preferences to optimize the list holistically. Generator-evaluator methods Huzhang et al. (2023); Shi et al. (2023) first generate multiple candidate lists and then evaluate them using a learned utility function to select the best one. They can effectively generate a high-quality recommendation list but suffer from the increasing computational cost due to the generation-evaluation loop. Emerging techniques leverage diffusion models Mao et al. (2025c;a;d) to generate refined item lists by iteratively denoising a ranking distribution, which excels in capturing complex user preferences.

### 2.3 ADVERSARIAL LEARNING

Adversarial learning Lowd & Meek (2005) has emerged as a powerful technique for data augmentation in recommendation systems, improving the model robustness and generalization. By leveraging generative adversarial networks (GANs) Wang et al. (2017) or adversarial training He et al. (2018); Tang et al. (2020), these methods synthesize high-quality user-item interactions or perturb existing data to enhance model robustness. For instance, IRGAN Wang et al. (2017) employs a minimax game between a generator (sampling hard negative items) and a discriminator (distinguishing real from synthetic interactions) to improve ranking performance. Similarly, AdvIR Park & Chang (2019) introduces adversarial perturbations to embedding spaces, forcing the model to learn more generalizable representations. These methods demonstrate that adversarial learning can not only augment training data but also enhance models' robustness. Our method is designed to add noise by adversarial learning, which can enhance the reranker's effectiveness in denoising the pre-ranking scores to align with user feedback.

## 3 METHODOLOGY

### 3.1 PROBLEM FORMULATION

For a given user request $\mathbf{u}$ (which may contain information about the candidate items $\mathcal{I}(u) = \{i_1, i_2, \cdots, i_n\}$ retrieved by the retriever, user profile features, and user interaction history), we

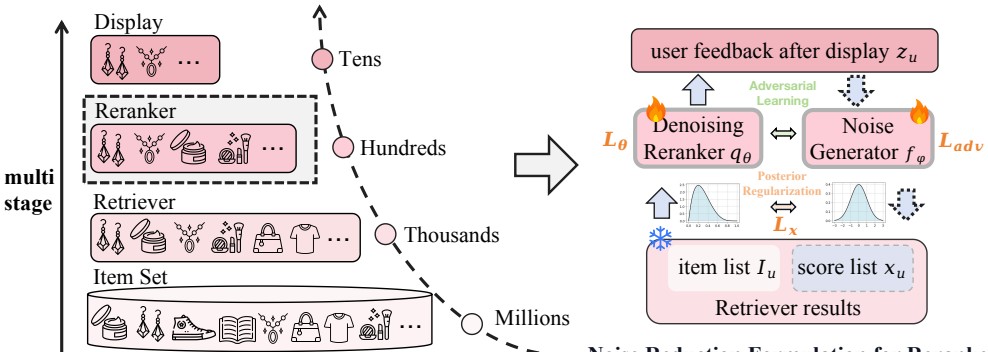

Figure 2: Overall framework of multi-stage recommender system (on the left) and the noise reduction formulation of our method, DNR.

observe the following during data collection: 1) The corresponding (continuous) retriever scores of $\mathcal{I}(u)$ are $\mathbf{x}_u = [x_{i_1}, x_{i_2}, \cdots, x_{i_n}] \in [0, 1]^n$, which essentially describe the user preference prediction in the view of the retriever; 2) The ground truth user feedback (*i.e.,* binary labels such as click, watch, share, etc.) of the candidate set, *i.e.,* $\mathbf{z}_u = [z_{i_1}, z_{i_2}, \cdots, z_{i_n}] \in \{0, 1\}^n$. Note that for recommender systems that further select K items after the reranking process, we may never observe the positive user feedback for the remaining $n - K$ items. From a probabilistic viewpoint, we can assume zero labels as defaults for these items, since passing the reranking process is the premise of observing the user feedback signals.

**The Noise Reduction Task:** In our formulation, the overall recommendation-feedback process $\mathbf{u} \rightarrow \mathbf{x} \rightarrow \mathbf{z}$ (where $\mathbf{x}$ and $\mathbf{z}$ are random variables for retriever scores and user feedback) involves an initial retriever $p_{\mathrm{x}}(\mathbf{x}|\mathbf{u})$, which serves as the retriever score generation prior (and is assumed non-optimizable in the reranking task). The learnable reranker $q_\theta(\mathbf{z}|\mathbf{x}, \mathbf{u})$ serves as the conditional user feedback likelihood estimator that predicts the user feedback $\hat{\mathbf{z}}$ given the retriever scores $\mathbf{x}$ and the user context $\mathbf{u}$, *i.e.,* $\hat{\mathbf{z}} \sim q_\theta(\cdot|\mathbf{x}_u, \mathbf{u})$. Given an observed feedback $\mathbf{z}_u$, the learning goal adopts the general data log likelihood maximization objective:

$$\max_\theta \log p(\mathbf{z}_u|\mathbf{u}, \theta) \tag{1}$$

where the predicted likelihood $p(\mathbf{z}_u|\mathbf{u}, \theta)$ for the observed user feedback $\mathbf{z}_u$ is determined by the retriever $p_{\mathrm{x}}$, the reranker $q_\theta$, and the user request $\mathbf{u}$.

**Note:** For simplicity of the presentation, we omit the user condition term $\mathbf{u}$ for the remainder of the paper (*e.g.,* we will denote the retriever as $p_{\mathrm{x}}(\mathbf{x})$, the reranker as $q_\theta(\mathbf{z}|\mathbf{x}_u)$, and simplify the learning goal as $\max_\theta \log p(\mathbf{z}_u)$), since we adopt an analysis framework on the user request level.

## 3.2 ANALYTICAL LIMITATION OF DIRECT OPTIMIZATION METHODS

As we have mentioned in section 1, involving retrieval scores as input is beneficial for existing reranking methods. Regardless of the reranker design, the corresponding objective function is:

$$\mathcal{L}_{\mathrm{direct}} = -\mathbb{E}_{\mathbf{x} \sim p_{\mathrm{x}}}[\log q_\theta(\mathbf{z}_u|\mathbf{x})] \tag{2}$$

which maximizes the user behavior alignment conditioned on the given retrieval scores. As a special case, according to the binary nature of $\mathbf{z}_u$, we can simplify the learning objective as standard binary cross-entropy (BCE) loss. Theoretically, we can find that $\mathcal{L}_{\mathrm{direct}}$ optimizes an upper bound of the negative data log likelihood (details in Appendix B.2):

$$-\log p(\mathbf{z}_u) = \mathcal{L}_{\mathrm{direct}} + L_1 + L_2$$
$$L_1 = \mathbb{E}_{\mathbf{x} \sim p_{\mathrm{x}}}\left[\log \frac{q_\theta(\mathbf{z}_u|\mathbf{x})}{p_{\mathrm{z|x}}(\mathbf{z}_u|\mathbf{x})}\right] \tag{3}$$
$$L_2 = -D_{\mathrm{KL}}\big(p_{\mathrm{x}}(\mathbf{x})\|p_{\mathrm{x|z}}(\mathbf{x}|\mathbf{z}_u)\big).$$

where $p_{z|x}$ represents the ground-truth user feedback probability and $p_{x|z}$ represents the posterior distribution of retriever scores conditioned on user feedback, both terms are not related to the reranker $q_\theta$, and they are merely determined by the retriever prior and the user. Intuitively, smaller values of $L_1$ and $L_2$ make the solution of Eq.(2) more aligned with the goal of Eq.(1). However, these two terms are not fully controlled by the direct optimization framework, potentially causing significant misalignment between Eq.(2) and Eq.(1). This will lead to a discrepancy between the final reranking results and user feedback, and thus, the suboptimal recommendation performance.

### 3.3 IMPLEMENTING NOISE GENERATOR FOR THE POSTERIOR

To overcome the aforementioned limitations, we propose to complement the denoising reranker $q_\theta$ with a noise generator $f_\phi(\cdot|\mathbf{z}_u)$. Formally, we generate noise with $\boldsymbol{\epsilon} \sim f_\phi$ and add it to the user feedback to form the synthetic retriever score posterior $p_\phi(\mathbf{x}|\mathbf{z}_u)$, taking insight from the reparameterization trick Kingma & Welling (2014):

$$\mathbf{x}'_u = (1 - \lambda_c)\mathbf{z}_u + \lambda_c\boldsymbol{\epsilon}, \tag{4}$$

where the generated noise $\boldsymbol{\epsilon}$ represents the noisy behavior in retriever scores $\mathbf{x}'_u$, $\lambda_c$ represents the proportion of noise in the generated retriever scores. We denote the generation process as $\mathbf{x}'_u \sim p_\phi$, which is analogous to the sampling process with ground truth posterior (assumed unknown and intractable), i.e., $\mathbf{x} \sim p_{x|z}$. For implementation, we investigate two choices of noise generator $f_\phi$:

- **Heuristic Generator:** This option uses a non-parametric noise generator $f_\phi^{\text{heuristic}}$ with a manually designed distribution. For example, we can adopt random Gaussian noise similar to Ho et al. (2020); Li et al. (2024), but we notice that this is not a good representation of the noise in retriever scores. Intuitively, given the binary user feedback signals, a natural assumption of the retriever prior is the Beta distribution, *i.e.,* , the conjugate prior and the posterior for binary variables (i.e., bernoulli or binomial distribution) are both beta distribution, where we can derive the heuristic solution as $\boldsymbol{\epsilon} \sim \text{Beta}(\alpha, \beta)$. For simplicity, one may set $\alpha = \beta = 0.5$ to approximate the uncertainty. In practice, we find that it is also beneficial to set $\alpha$ and $\beta$ that align with the observed noise distribution in the data.

- **Model-based Generator:** While the heuristic generator is effective in our empirical study, we are skeptical of whether the manually designed heuristic noise generator is optimal for augmenting the retriever score distribution. Since the prior noise distribution is uncertain, it is necessary to learn this distribution adaptively and personally. To achieve a more precise estimation, we design a learnable model $f_\phi^{\text{model}}$, with parameter $\phi$, that will be optimized along with the reranker (details in the next section). The generated noise $\boldsymbol{\epsilon}$ should also be aligned with the observed distribution, and it also participates in an adversarial training objective in order to better explore the retriever score space, which in turn improves the reranker's effectiveness. As the noise is learned, $\lambda_c$ requires no manual adjustment and can be treated as 1. As we will show in Section 4.3, this choice is empirically better than the heuristic generator.

### 3.4 OVERALL LEARNING FRAMEWORK

In this section, we show how to optimize the user feedback alignment goal, *i.e.,* Eq.(1) by simultaneously learning the denoising reranker and the adversarial noise generator. Specifically, we can decompose the negative log-likelihood objective into three loss terms (derivation in Appendix B.1):

$$-\log p(\mathbf{z}_u) = \mathcal{L}_z + \mathcal{L}_{\text{adv}} + \mathcal{L}_x + \delta_x$$
$$\mathcal{L}_z = -\mathbb{E}_{\mathbf{x}\sim p_\phi}\Big[\log q_\theta(\mathbf{z}_u|\mathbf{x})\Big]$$
$$\mathcal{L}_{\text{adv}} = \mathbb{E}_{\mathbf{x}\sim p_\phi}\Big[\log \frac{q_\theta(\mathbf{z}_u|\mathbf{x})}{p_{z|x}(\mathbf{z}_u|\mathbf{x})}\Big] \tag{5}$$
$$\mathcal{L}_x = D_{\text{KL}}\Big(p_\phi(\mathbf{x}|\mathbf{z}_u)\|p_x(\mathbf{x})\Big),$$

which consists of an augmented denoising loss term $\mathcal{L}_z$, an adversarial noise generation loss $\mathcal{L}_{\text{adv}}$, and a score distribution regularization term $\mathcal{L}_x$. We will illustrate the details of these three terms in the following sections. The additional term $\delta_x = -D_{\text{KL}}\big(p_\phi(\mathbf{x}|\mathbf{z}_u)\|p_{x|z}(\mathbf{x}|\mathbf{z}_u)\big) \leq 0$ represents

the negative divergence between the generated noisy retriever scores $p_\phi$ and the corresponding real $p_{x|z}$. This term is not directly optimizable due to the unknown $p_{x|z}$, but we expect to achieve a small divergence in $\delta_x$ with the optimization of the other three terms, so that the overall optimization is aligned with the goal of Eq.(1).

**Denoising with Augmented Retriever Scores:** $\mathcal{L}_z$ in Eq.(5) denotes the denoising loss under the noisy retriever scores generated by the noise generator. Specifically, we can first synthesize $\mathbf{x}'_u \sim p_\phi(\cdot|\mathbf{z}_u)$ as the generated retriever scores according to Eq.(4), then pass them into the reranker $q_\theta$ for the prediction of user feedback $\mathbf{z}_u$, and $\mathcal{L}_z$ only updates $\theta$. The optimization can also be simplified into the BCE loss with $\mathbf{z}_u$ as labels. Note that the generated scores $\mathbf{x}'_u$, different from the observed $\mathbf{x}_u$, essentially serve as the augmented scores that the initial retriever would have produced given the user request. This means that the reranker will be aware of the noisy behavior of the retriever. Combining with the direct optimization, we have the final denoising loss for the reranker:

$$\mathcal{L}_\theta = \mathcal{L}_{\text{direct}} + \lambda_m \mathcal{L}_z, \tag{6}$$

where $\lambda_m$ controls the magnitude of the augmentation, and the parameters of the noise generator $f_\phi$ are fixed. Additionally, our model-agnostic framework does not restrict the architecture of the denoising reranker $q_\theta$, as we will show in section 4.3, one can adopt various backbones, including PRM Pei et al. (2019) and PIER Shi et al. (2023).

**Adversarial Noise Learning:** $\mathcal{L}_{\text{adv}}$ describes how the reranker $q_\theta$ behaves differently from the user feedback under the viewpoint of the synthetic retriever scores. In practice, this term is expected to be small since the ratio $q_\theta(\mathbf{z}_u|\mathbf{x})/p_{z|x}(\mathbf{z}_u|\mathbf{x}) \approx 1$ in major cases due to the user feedback alignment of Eq.(6) and the augmented objective $\mathcal{L}_z$. Nevertheless, in the view of generated noisy retriever scores, we can see that $\mathcal{L}_{\text{adv}}$ encourages adversarial samples of $\mathbf{x}$ that produce small ratios of $q_\theta(\mathbf{z}_u|\mathbf{x})/p_{z|x}(\mathbf{z}_u|\mathbf{x})$. In other words, a better $p_\phi$ should be able to explore and synthesize retriever scores that are incorrectly denoised by the reranker $q_\theta$. For the heuristic noise generator $f_\phi^{\text{heuristic}}$, it is not optimizable once $\alpha$ and $\beta$ are chosen. In contrast, for the model-based generator $f_\phi^{\text{model}}$, we follow the intuition of adversarial case exploration, and approximate $\mathcal{L}_{\text{adv}}$ by first fixing the reranker and then optimizing the generator with an adversarial loss:

$$\min_\phi \log q_\theta(\mathbf{z}_u|\mathbf{x}'_u). \tag{7}$$

We implement the model-based generator $f_\phi^{\text{model}}$ by a 2-layer MLP and directly output $\mathbf{x}'_u$ using Eq.(4), so that the end-to-end learning becomes feasible for Eq.(7). In cases where the noise generator is not directly optimizable, one may have to consider other alternatives like reparameterization tricks Kingma & Welling (2014) or policy gradient Silver et al. (2014). Notably, $\mathcal{L}_{\text{adv}}$ does not encourage over-exploration of $\mathbf{x}'_u$, which may negatively impact the learning of the reranker. In such cases, the synthetic scores are unrealistically different from the real retriever scores, diverging from the user feedback, inducing small $p_{z|x}$ and large $q_\theta(\mathbf{z}_u|\mathbf{x})/p_{z|x}(\mathbf{z}_u|\mathbf{x})$, violating $\mathcal{L}_{\text{adv}}$.

**Noise Regularization:** $\mathcal{L}_x$ aims to reduce the divergence between the synthetic retriever score distribution $p_\phi$ and the real retriever score prior distribution $p_x$. Intuitively, this term favors the synthetic score distribution $p_\phi$ that can mimic the noisy behavior of the real retriever's scores $p_x$. For implementation, we first synthesize the retriever score $\mathbf{x}'_u$ with Eq.(4) and calculate the distributional difference between synthetic scores from $p_\phi$ and retriever scores from $p_x$. Then, we minimize this difference to approximate the distribution-level KL divergence.

Again, given that the heuristic noise generator $f_\phi^{\text{heuristic}}$ is not optimizable by the objectives in Eq (5), it might not be the optimal choice for augmenting the retriever score distribution. As a consequence, this could lead to recommendations that are not in alignment with user feedback. For the training of the model-based noise generator $f_\phi^{\text{model}}$, we first adopt the heuristic generator in the first $\lambda_e$ epochs to ensure learning stability of reranker $q_\theta$ throughout $\mathcal{L}_z$, then engage the adversarial learning of $\mathcal{L}_{\text{adv}}$ and the optimization of $\mathcal{L}_x$ afterward, switching the generation of $\epsilon$ from heuristic generator to model-based generator. We present the algorithm of our training procedure in Appendix C.

## 4 EXPERIMENTS

In this section, we conduct extensive experiments to answer the following questions:

- RQ1: How does DNR perform compared with leading rerankers in recommendation systems?

- RQ2: How does the denoising formulation compare to other methods of using retriever scores?

- RQ3: Can the denoising formulation generalize to reranker backbones other than PRM?

- RQ4: How does the heuristic noise generator's performance compare to that of the adversarial model-based generator?

- RQ5: How does different objectives in DNR contribute to the performance?

- RQ6: How sensitive is DNR to the hyperparameters $\lambda_c$, $\lambda_m$, and $\lambda_e$?

## 4.1 EXPERMENTAL SETTINGS

**Datasets.** We conduct experiments on three real-world datasets, including ML-1M Harper & Konstan (2016), Kuaivideo Liu et al. (2025), and Amazon-books Ni et al. (2019). The detailed description and the statistics of the processed datasets for both the retriever and the reranker stages are presented in Appendix D.1. For the online experimental verification, please refer to Appendix E.6.

**Baselines:** We compare the performance of DNR with four categories of recommendation methods in our reranking task, including traditional recommenders (*i.e.,* SASRec Kang & McAuley (2018), Caser Tang & Wang (2018), GRU4Rec Hidasi et al. (2016), and MiDNN Zhuang et al. (2018)), which individually predict the scores of candidate items and rank them accordingly; **list-refinement methods** (*i.e.,* DLCM Ai et al. (2018), SetRank Pang et al. (2020), PRM Pei et al. (2019), and MIR Xi et al. (2022)), which encode the mutual inference in candidate items and refine the rankings based on this information; **generator-evaluator methods** (*i.e.,* EGRerank Huzhang et al. (2023), Pier Shi et al. (2023), and NAR4Rec Ren et al. (2024)), which generate multiple reranked item lists and evaluate them to select the best one for users; and **diffusion-Based methods** (*i.e.,* DiffuRec Li et al. (2024) and DCDR Lin et al. (2024)) which utilize diffusion models to generate items or lists for recommendation. Detailed information for our baselines is presented in Appendix D.3.

**Implementation Details:** Our experiments are implemented with Python 3.8 and PyTorch 1.12. We construct the retriever stage with the collaborative filtering method Koren et al. (2009) to obtain the initial retriever scores for the top-50 candidate items. The detailed setting and experimental results of the retriever stage are presented in Appendix E.1. The reranker reorders these items filtered by retrievers and selects the top $K = 6$ for display. The dimension of rerankers' hidden embedding is 128 across all models, and all models are trained until convergence. The learning rate and weight decay are tuned within the range of $[0.01, 0.001, 0.0001]$ and $[0, 0.1, 0.01, 0.001]$, respectively. We tune the hyperparameters with $\lambda_c \in [0.1, 1.0]$, $\lambda_m \in [0.1, 1.0]$, and $\lambda_e \in [0, 200]$. The optimal hyperpapemters adopted for our experimental results are presented in Appendix D.2.

**Evaluation Metrics:** Following common practices, we employ Hit Ratio@6 (H@6), NDCG@6 (N@6), MAP@6 (M@6), F1@6, and AUC for performance evaluation of the reranker stage. Although not the focus of this paper, we adopt the AUC metric to evaluate the performance of the retriever stage. And the retriever is fixed during training of rerankers.

## 4.2 MAIN RESULTS (RQ1)

We compare the performance of our proposed DNR with multiple leading baselines and the results are detailed in Table 1. "DNR-G" and "DNR-B" denote using two alternatives of $f_\phi^{\text{heuristic}}$ to generate Gaussian noise and Beta noise, respectively, before switching to the model-based noise generator $f_\phi^{\text{model}}$ in the epoch $\lambda_e$. We can observe that the generator-evaluator-based methods (*i.e.,* EGRerank Huzhang et al. (2023), Pier Shi et al. (2023), and NAR4Rec Ren et al. (2024)) achieve comparable performance among the multiple baselines, validating the effectiveness of the generator-evaluator paradigm in the reranker stage. In comparison, our DNR solutions consistently outperform the state-of-the-art baselines across the three benchmarks. This highlights the superiority of our noise reduction formulation and the significance of improving reranker effectiveness with three objectives in adversarial learning.

Table 1: Overall performance of different methods for the reranking recommendation. The best scores are in bold and the best baseline scores are underlined, respectively. All improvements are statistically significant with student t-test $p < 0.05$.

| Methods | ML-1M | | | | | Kuaivideo | | | | | Book | | | | |
|---|---|---|---|---|---|---|---|---|---|---|---|---|---|---|---|
| | H@6 | N@6 | M@6 | F1@6 | AUC | H@6 | N@6 | M@6 | F1@6 | AUC | H@6 | N@6 | M@6 | F1@6 | AUC |
| SASRec | 59.79 | 72.16 | 61.84 | 65.41 | 85.98 | 36.72 | 54.01 | 41.66 | 43.00 | 77.86 | 60.27 | 69.32 | 58.17 | 62.44 | 83.64 |
| Caser | 58.60 | 71.34 | 60.55 | 64.14 | 86.53 | 36.66 | 54.04 | 41.65 | 43.93 | 78.39 | 59.92 | 69.31 | 58.06 | 62.09 | 84.21 |
| GRU4Rec | 58.07 | 69.91 | 59.03 | 63.47 | 86.74 | 37.28 | 54.39 | 42.20 | 43.66 | 74.67 | 53.69 | 59.24 | 46.71 | 55.65 | 81.19 |
| MiDNN | 56.86 | 70.30 | 59.28 | 62.16 | 86.87 | 37.32 | 54.56 | 42.38 | 43.72 | 74.73 | 60.28 | 69.61 | 58.58 | 62.45 | 83.02 |
| DLCM | 62.31 | 73.87 | 63.82 | 67.96 | 89.35 | 39.69 | 60.67 | 48.90 | 46.61 | 75.80 | 66.80 | 75.88 | 65.39 | 69.28 | 91.93 |
| SetRank | 59.35 | 73.10 | 62.51 | 64.72 | 88.84 | 44.75 | 64.39 | 51.88 | 52.59 | 89.93 | 66.15 | 75.70 | 64.84 | 68.64 | 92.01 |
| PRM | 60.09 | 72.85 | 62.21 | 65.51 | 88.20 | 39.92 | 55.93 | 42.97 | 46.18 | 85.15 | 67.86 | 76.88 | 66.44 | 70.42 | 92.00 |
| MIR | 62.22 | 74.33 | 64.47 | 67.97 | 87.76 | 37.01 | 55.79 | 43.16 | 44.50 | 79.95 | 66.08 | 71.48 | 56.62 | 68.62 | 91.82 |
| EGRank | 62.76 | 74.75 | 64.97 | 68.46 | 88.72 | 40.09 | 59.01 | 47.52 | 47.06 | 77.44 | 70.73 | 80.75 | 72.40 | 73.33 | 89.40 |
| Pier | 62.74 | 75.99 | 65.98 | 68.74 | 90.43 | 45.35 | 65.11 | 52.55 | 53.35 | 90.93 | 71.14 | 80.22 | 71.62 | 73.74 | 92.26 |
| NAR4Rec | 62.81 | 75.01 | 65.42 | 68.31 | 88.30 | 44.31 | 63.83 | 51.45 | 52.08 | 89.94 | 70.08 | 79.46 | 70.69 | 72.66 | 92.44 |
| MG-E | 63.53 | 76.18 | 66.52 | 69.37 | 89.08 | 46.89 | 66.19 | 53.75 | 55.26 | 90.16 | 70.63 | 79.81 | 71.07 | 73.20 | 90.21 |
| DiffuRec | 62.16 | 75.80 | 66.28 | 67.83 | 91.48 | 40.78 | 61.16 | 50.77 | 48.00 | 81.08 | 58.62 | 68.77 | 57.25 | 60.79 | 84.74 |
| DCDR | 60.79 | 73.21 | 62.57 | 66.58 | 89.01 | 43.86 | 63.52 | 50.92 | 51.66 | 89.92 | 65.84 | 70.36 | 54.76 | 67.33 | 88.07 |
| DNR-G | **64.89↑** | **77.67↑** | **68.00↑** | **70.71↑** | **92.75↑** | 49.61↑ | 69.60↑ | 57.67↑ | 58.39↑ | 93.20↑ | 74.83↑ | 82.57↑ | 74.24↑ | 77.65↑ | **94.46↑** |
| DNR-B | 64.23↑ | 77.12↑ | 67.30↑ | 70.01↑ | 92.08↑ | **50.30↑** | **70.15↑** | **58.12↑** | **59.16↑** | **93.38↑** | **75.50↑** | **83.53↑** | **75.54↑** | **78.30↑** | 94.23↑ |

## 4.3 ABLATION STUDY (RQ2, RQ3, RQ4, & RQ5)

**RQ2 & RQ3:** To investigate different ways to leverage retriever scores, we develop three variants for leveraging retriever scores as rerankers' input: "c score" refers to leveraging a concatenate operation to combine the score features with encoded user states from users' interaction history; "+ score" denotes incorporating the score feature with user states by a simple addition; "w score" represents adopting the initial retriever scores as the weights of candidate items. The frameworks of the three variants are shown in Appendix D.4. To validate the generalizability of DNR across different reranker backbones, we conduct experiments comparing our three variants and DNR itself when using PRM Pei et al. (2019) and Pier Shi et al. (2023) as the underlying rerankers, respectively. We report the experimental results in Table 2. The variants "c score", "+ score", and "w score" all achieve superior performance compared to the base reranker backbone, confirming that incorporating retriever scores into the reranking stage yields tangible benefits. Moreover, our DNR outperforms all three variants, underscoring denoising formulation as a more effective strategy for leveraging retriever scores. Importantly, DNR delivers consistent improvements across both backbones, thereby validating the generalizability of our denoising formulation for reranking architectures.

**RQ4:** To further investigate different noise generators, we conduct ablation experiments on the heuristic noise generators that generate noise from the Gaussian distribution and Beta distribution, denoted as "w/ G" and "w/ B" respectively. The results are presented in Table 3. By comparing "w/ G" with "DNR-G or comparing "w/ B" with "DNR-B", we can see that our DNR framework—equipped with a model-based noise generator for adversarial learning—consistently outperforms counterparts using heuristic noise generators.

**RQ5:** To compare different learning objectives in Eq.(5), we first observe that "w/ G" and "w/B" methods outperform the PRM baseline, indicating the effectiveness of $\mathcal{L}_z$, which uses the heuristic noise generator to augment the learning of reranker. Then, all "w/o $\mathcal{L}_{adv}$" alternatives and "w/o $\mathcal{L}_x$" alternatives appear to downgrade the effectiveness of the full implementation of DNR. All these observations validate the effectiveness of the derived terms in Eq.(5).

Table 2: Ablation Study for the different ways of leveraging the retriever scores on both PRM and Pier reranker backbones.

| Methods | ML-1M | | | | | Kuaivideo | | | | | Book | | | | |
|---|---|---|---|---|---|---|---|---|---|---|---|---|---|---|---|
| | H@6 | N@6 | M@6 | F1@6 | AUC | H@6 | N@6 | M@6 | F1@6 | AUC | H@6 | N@6 | M@6 | F1@6 | AUC |
| PRM | 60.09 | 72.85 | 62.21 | 65.51 | 88.20 | 39.92 | 55.93 | 42.97 | 46.18 | 85.15 | 67.86 | 76.88 | 66.44 | 70.42 | 92.00 |
| c score | 62.50↑ | 76.73↑ | 67.36↑ | 68.29 | 91.35↑ | 46.20↑ | 67.58↑ | 56.31↑ | 54.25↑ | 85.76↑ | 69.71↑ | 77.99↑ | 68.20↑ | 72.29↑ | 92.15↑ |
| + score | 62.69↑ | 76.05↑ | 66.48↑ | 68.36↑ | 90.77↑ | 45.83↑ | 66.77↑ | 56.25↑ | 53.97↑ | 83.47 | 71.50↑ | 81.18↑ | 72.72↑ | 74.18↑ | 91.41 |
| w score | 62.48↑ | 76.68↑ | 67.34↑ | 68.19↑ | 87.20 | 46.82↑ | 66.94↑ | 54.77↑ | 55.00↑ | 84.05 | 72.73↑ | 82.21↑ | 74.15↑ | 75.45↑ | 91.81 |
| DNR | 64.23 | 77.12 | 67.30 | 70.01 | 92.08 | 50.30 | 70.15 | 58.12 | 59.16 | 93.38 | 75.50 | 83.53 | 75.54 | 78.30 | 94.23 |
| Pier | 62.74 | 75.99 | 65.98 | 68.74 | 90.43 | 45.35 | 65.11 | 52.55 | 53.35 | 90.93 | 71.14 | 80.22 | 71.62 | 73.74 | 92.26 |
| c score | 62.33 | 76.51↑ | 66.95↑ | 69.18↑ | 90.55↑ | 46.28↑ | 66.09↑ | 53.98↑ | 53.76↑ | 90.56 | 71.88↑ | 81.02↑ | 73.15↑ | 74.32↑ | 91.89 |
| + score | 62.58 | 76.18↑ | 66.52↑ | 68.85↑ | 90.31 | 46.83↑ | 68.21↑ | 55.88↑ | 54.66↑ | 91.82 | 71.27↑ | 80.70↑ | 72.79↑ | 74.15↑ | 92.28↑ |
| w score | 62.98↑ | 76.66↑ | 67.16↑ | 69.37↑ | 90.58↑ | 45.76↑ | 66.82↑ | 54.55↑ | 53.33 | 90.45 | 72.03↑ | 81.56↑ | 74.74↑ | 75.32↑ | 92.33↑ |
| DNR | 63.41 | 77.28 | 68.02 | 70.22 | 91.97 | 48.89 | 69.35 | 57.76 | 56.02 | 92.38 | 73.68 | 82.61 | 74.82 | 76.88 | 93.78 |

Table 3: Ablation Study for DNR alternatives. "G w/o $\mathcal{L}_{adv}$" and "B w/o $\mathcal{L}_{adv}$" represent DNR-G or DNR-B paradigms without adversarial noise learning objective; "G w/o $\mathcal{L}_x$" and "B w/o $\mathcal{L}_x$" refer to DNR-G or DNR-B paradigms without the noise regularization.

| Methods | ML-1M | | | | | Kuaivideo | | | | | Book | | | | |
|---|---|---|---|---|---|---|---|---|---|---|---|---|---|---|---|
| | H@6 | N@6 | M@6 | F1@6 | AUC | H@6 | N@6 | M@6 | F1@6 | AUC | H@6 | N@6 | M@6 | F1@6 | AUC |
| PRM | 60.09 | 72.85 | 62.21 | 65.51 | 88.20 | 39.92 | 55.93 | 42.97 | 46.18 | 85.15 | 67.86 | 76.88 | 66.44 | 70.42 | 92.00 |
| w/ G | 63.36 | 76.69 | 66.82 | 69.34 | 91.98 | 47.84 | 67.34 | 55.27 | 56.28 | 92.43 | 73.38 | 81.32 | 72.89 | 76.18 | 93.38 |
| w/ B | 63.65 | 76.90 | 67.01 | 69.64 | 90.85 | 48.83 | 68.71 | 56.60 | 57.45 | 92.73 | 73.42 | 81.20 | 72.74 | 76.23 | 93.46 |
| G w/o $\mathcal{L}_{adv}$ | 63.71 | 76.57 | 66.61 | 69.43 | 92.14 | 48.75 | 68.70 | 56.57 | 57.34 | 92.21 | 73.83 | 81.89 | 73.27 | 76.60 | 93.62 |
| G w/o $\mathcal{L}_x$ | 63.47 | 76.43 | 66.43 | 69.19 | 91.90 | 48.88 | 69.10 | 57.18 | 57.57 | 92.69 | 73.99 | 82.01 | 73.44 | 76.76 | 94.29 |
| DNR-G | 64.89 | 77.67 | 68.0 | 70.71 | 92.75 | 49.61 | 69.60 | 57.67 | 58.39 | 93.20 | 74.83 | 82.57 | 74.24 | 77.65 | 94.46 |
| B w/o $\mathcal{L}_{adv}$ | 63.65 | 76.90 | 67.01 | 69.64 | 90.85 | 49.55 | 69.84 | 58.04 | 58.31 | 92.91 | 74.71 | 82.49 | 74.14 | 77.52 | 94.01 |
| B w/o $\mathcal{L}_x$ | 63.32 | 76.62 | 66.76 | 69.29 | 90.60 | 49.96 | 69.93 | 58.00 | 58.77 | 92.81 | 74.42 | 82.20 | 73.74 | 77.23 | 93.96 |
| DNR-B | 64.23 | 77.12 | 67.30 | 70.01 | 92.08 | 50.30 | 70.15 | 58.12 | 59.16 | 93.38 | 75.50 | 83.53 | 75.54 | 78.30 | 94.23 |

## 4.4 SENSITIVITY ANALYSIS (RQ6)

We conduct sensitivity analysis for $\lambda_c$, $\lambda_m$, and $\lambda_e$ for both DNR-G and DNR-B, and present the results in Figure 3. When adjusting $\lambda_m$, $\lambda_e$ is fixed at 40; when adjusting $\lambda_e$, $\lambda_m$ is fixed at 0.3. For $\lambda_c$ adjustments, $\lambda_m$ and $\lambda_e$ are set to 0.6 and 80, respectively. While the Beta noise generally performs better than Gaussian noise in most settings, as shown in Table 2, there exist some common patterns in these hyper-parameters: 1) all parameters appears to have an optimal value within the searched region, indicating the effectiveness of including each corresponding designs, *i.e.*, $\lambda_c$ for the synthetic noisy scores in Eq.(4), $\lambda_m$ for the combined denoising loss Eq.(6), and $\lambda_e$ for the switching from heuristic noise generator to model-based one; 2) Over-amplifying these hyper-parameters results in the downgrade of the rerank performance, indicating the significance of keeping a balance between the involved components. Additionally, the analysis of $\lambda_c$ in the left plot of Figure 3 shows that using the model-based noise generator can further enhance the performance over the heuristic one.

## 5 CONCLUSION

In this paper, we provide evidence that the initial retriever scores are informative for the latter reranker stage in the multi-stage recommender system. To formulate the relationship between retriever and reranker scores, we view the reranker as a noise reduction task from the retriever scores and

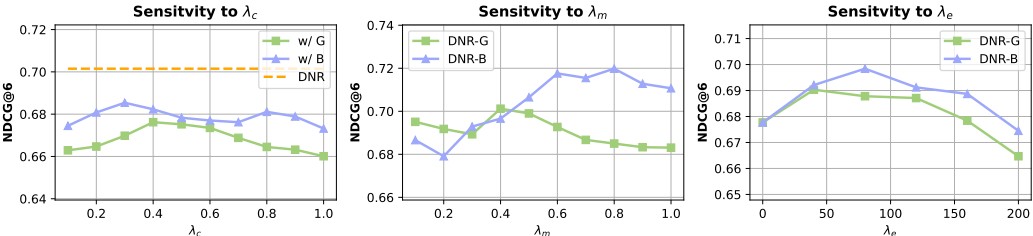

Figure 3: Sensitivity analysis of hyperparameters (*i.e.,* $\lambda_c$, $\lambda_m$, and $\lambda_e$) for our method on the Kuaivideo dataset.

propose DNR. To fully explore the score space, we propose the framework of DNR, adding noise to generate retriever scores as augmentation and then denoising them to align with user feedback. While a heuristic design of the noise distribution can effectively improve the model performance (*e.g.,* Gaussian or Beta distribution), we prove that adversarially learns a denoising reranker and a noise generator for the retriever stage can further boost the reranker's performance. Theoretical analysis and empirical validation across three public datasets demonstrate the effectiveness of the noise reduction formulation and the design of our DNR.

## 6 ACKNOWLEDGEMENT

This research was supported by the National Natural Science Foundation of China (U25A20445) and the advanced computing resources provided by the Supercomputing Center of the University of Science and Technology of China (USTC). This work was also supported by Kuaishou Technology, where author Wenyu Mao conducted this research during an internship.

## ETHICS STATEMENT.

This work is designed to explore the significance of retriever scores for the reranker, formulating the reranker as a denoising process of retriever scores with an adversarial framework. We do not foresee any direct, immediate, or negative societal impacts of our research.

## REPRODUCIBILITY STATEMENT.

All the results in this work are reproducible. We have discussed the optimal hyperparameters and the details on devices and software environments in Section 4.1 and Appendix D.2. For reproducibility, we release the code at `https://github.com/maowenyu-11/DNR`

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

# A    NOTATIONS

| symbol | description |
|---|---|
| $\mathbf{u}$ | user request information including profile features and interaction history |
| $\mathbf{x}$ | random variables of retriever scores on the candidate item set |
| $\mathbf{z}$ | random variables of user feedback on the candidate item set |
| $\mathbf{x}_u, \mathbf{z}_u$ | observed retriever scores and user feedback in data |
| $\mathbf{x}'_u$ | the generated retriever scores from the learned or heuristic posterior model $\phi$ |
| $q_\theta$ | the probabilistic representation of the reranker model |
| $p_\phi$ | the probabilistic representation of the posterior |
| $f_\phi$ | the noise generator that implements the posterior |
| $p_{\mathrm{x}}$ | the probabilistic representation of the non-optimizable retriever |
| $p_{\mathrm{x|z}}$ | the posterior of a given reranker |
| $p_{\mathrm{z|x}}$ | the probabilistic representation of ground truth user feedback |
| $\mathcal{L}_{\mathrm{direct}}$ | the standard user feedback alignment loss given the retriever scores |
| $\mathcal{L}_{\mathrm{z}}$ | the user feedback alignment loss given generated retriever scores from posterior |
| $\mathcal{L}_{\mathrm{adv}}$ | the adversarial loss that encourages the posterior to generate irregular noises |
| $\mathcal{L}_{\mathrm{x}}$ | the regularization of posterior |
| $\lambda_c$ | the hyperparameter that controls the magnitude of noise injection in Eq.(4) |
| $\lambda_m$ | the hyperparameter that balances $\mathcal{L}_{\mathrm{direct}}$ and $\mathcal{L}_{\mathrm{z}}$ when learning the reranker |
| $\lambda_e$ | the hyperparameter that determines which epoch switches the noise generator |

Table 4: List of Notations

# B    PROOFS

## B.1    DERIVATION OF THE THREE OBJECTIVES IN OUR SOLUTION

In this section, we illustrate the derivation of the Eq.(5). Similar to the variational Bayesian inference Kingma & Welling (2014), we start by analyzing the KL divergence between the real retriever scores and synthetic noisy scores:

$$
\begin{aligned}
D_{\mathrm{KL}}(p_\phi(\mathbf{x}|\mathbf{z}_u)\|p_{\mathrm{x|z}}(\mathbf{x}|\mathbf{z}_u)) &= -\mathbb{E}_{p_\phi}\left[\log\frac{p_{\mathrm{x|z}}(\mathbf{x}|\mathbf{z}_u)}{p_\phi(\mathbf{x}|\mathbf{z}_u)}\right] \\
&= -\mathbb{E}_{p_\phi}\left[\log\frac{p(\mathbf{x},\mathbf{z}_u)}{p_\phi(\mathbf{x}|\mathbf{z}_u)} - \log p(\mathbf{z}_u)\right] \qquad (8)\\
&= -\mathbb{E}_{p_\phi}\left[\frac{\log p(\mathbf{x},\mathbf{z}_u)}{\log p_\phi(\mathbf{x}|\mathbf{z}_u)}\right] + \log p(\mathbf{z}_u),
\end{aligned}
$$

Note that a key difference between Eq.(8) and the conventional derivation in Kingma & Welling (2014) is that the variational posterior $\phi$ is not a latent factor encoder that reduces noise and dimensions, but an explicit noise generator that mimics the behavior of retriever scores. In terms of the denoising problem formulation, it is also similar to a one-step diffusion, but with a key difference in that the real retriever scores $\mathbf{x}_u$ are not sampled by a manually defined distribution. Instead, an existing retriever $p_{\mathrm{x}}$ serves as the prior distribution.

Then, rearranging the terms of Eq.(8), we can derive the corresponding objectives that influence the data likelihood as follows:

$$
\begin{aligned}
& -\log p(\mathbf{z}_u) \\
=& -\mathbb{E}_{\mathbf{x}\sim p_\phi}\Big[\log\frac{p(\mathbf{x},\mathbf{z}_u)}{p_\phi(\mathbf{x}|\mathbf{z}_u)}\Big] - D_{\mathrm{KL}}\big(p_\phi(\mathbf{x}|\mathbf{z}_u)\|p_{\mathrm{x|z}}(\mathbf{x}|\mathbf{z}_u)\big) \\
=& -\mathbb{E}_{\mathbf{x}\sim p_\phi}\big[\log p_{\mathrm{z|x}}(\mathbf{z}_u|\mathbf{x})\big] + D_{\mathrm{KL}}\big(p_\phi(\mathbf{x}|\mathbf{z}_u)\|p_{\mathrm{x}}(\mathbf{x})\big) - D_{\mathrm{KL}}\big(p_\phi(\mathbf{x}|\mathbf{z}_u)\|p_{\mathrm{x|z}}(\mathbf{x}|\mathbf{z}_u)\big) \quad (9) \\
=& -\mathbb{E}_{\mathbf{x}\sim p_\phi}\big[\log q_\theta(\mathbf{z}_u|\mathbf{x})\big] + \mathbb{E}_{\mathbf{x}\sim p_\phi}\Big[\log\frac{q_\theta(\mathbf{z}_u|\mathbf{x})}{p_{\mathrm{z|x}}(\mathbf{z}_u|\mathbf{x})}\Big] + D_{\mathrm{KL}}\big(p_\phi(\mathbf{x}|\mathbf{z}_u)\|p_{\mathrm{x}}(\mathbf{x})\big) \\
& - D_{\mathrm{KL}}\big(p_\phi(\mathbf{x}|\mathbf{z}_u)\|p_{\mathrm{x|z}}(\mathbf{x}|\mathbf{z}_u)\big)
\end{aligned}
$$

which is equivalent to Eq.(5). It generally follows the derivation of the evidence lower bound in variational auto-encoders, except that we further decompose the first reconstruction term into the augmented denoising loss $\mathcal{L}_z = -\mathbb{E}_{\mathbf{x}\sim p_\phi}\big[\log q_\theta(\mathbf{z}_u|\mathbf{x})\big]$ and the adversarial learning loss $\mathcal{L}_{\mathrm{adv}} = \mathbb{E}_{\mathbf{x}\sim p_\phi}\big[\log\frac{q_\theta(\mathbf{z}_u|\mathbf{x})}{p_{\mathrm{z|x}}(\mathbf{z}_u|\mathbf{x})}\big]$. Intuitively, when $q_\theta$ is closely aligned with the real user feedback distribution $p_{\mathrm{z|x}}$, the adversarial term $\mathcal{L}_{\mathrm{adv}}$ will be close to zero. Minimizing $\mathcal{L}_{\mathrm{adv}}$ would encourage the noise generator $f_\phi$ to find samples where $q_\theta$ and $p_{\mathrm{z|x}}$ behave differently.

## B.2 THE ANALYTICAL LIMITATION OF STANDARD RERANKING MODELS

In this section, we formally analyze the Eq.(3). Recall that the direct optimization goal $\mathcal{L}_{\mathrm{direct}}|_{\mathbf{x}=\mathbf{x}_u} = -\mathbb{E}_{\mathbf{x}_u\sim p_{\mathrm{x}}}[\log q_\theta(\mathbf{z}_u|\mathbf{x}_u)]$, where the observed $\mathbf{x}_u$ only comes from the retriever prior $p_{\mathrm{x}}$ without posterior modeling. Note that we additionally define $p_{\mathrm{z|x}}$ as the ground-truth user feedback probability and $p_{\mathrm{x|z}}$ as the posterior distribution of retriever scores conditioned on user feedback. Both $p_{\mathrm{z|x}}$ and $p_{\mathrm{x|z}}$ are not related to the reranker $q_\theta$, and they are merely determined by the retriever prior and the user. Then, we can extract the user behavior alignment error of $q_\theta$ as follows and derive its relation to the goal of $\max_\theta \log p(\mathbf{z}_u)$:

$$
\begin{aligned}
-\mathbb{E}_{\mathbf{x}_u\sim p_{\mathrm{x}}}[\log q_\theta(\mathbf{z}_u|\mathbf{x}_u)] =& -\mathbb{E}_{\mathbf{x}_u\sim p_{\mathrm{x}}}\Big[\log\frac{q_\theta(\mathbf{z}_u|\mathbf{x}_u)}{p_{\mathrm{z|x}}(\mathbf{z}_u|\mathbf{x}_u)}\Big] - \mathbb{E}_{\mathbf{x}_u\sim p_{\mathrm{x}}}[\log p_{\mathrm{z|x}}(\mathbf{z}_u|\mathbf{x}_u)] \\
=& -\mathbb{E}_{\mathbf{x}_u\sim p_{\mathrm{x}}}\Big[\log\frac{q_\theta(\mathbf{z}_u|\mathbf{x}_u)}{p_{\mathrm{z|x}}(\mathbf{z}_u|\mathbf{x}_u)}\Big] + D_{\mathrm{KL}}\big(p_{\mathrm{x}}(\mathbf{x}_u)\|p_{\mathrm{x|z}}(\mathbf{x}_u|\mathbf{z}_u)\big) - \log p(\mathbf{z}_u) \\
\Rightarrow -\log p(\mathbf{z}_u) =& -\mathbb{E}_{\mathbf{x}_u\sim p_{\mathrm{x}}}[\log q_\theta(\mathbf{z}_u|\mathbf{x}_u)] + \mathbb{E}_{\mathbf{x}_u\sim p_{\mathrm{x}}}\Big[\log\frac{q_\theta(\mathbf{z}_u|\mathbf{x}_u)}{p_{\mathrm{z|x}}(\mathbf{z}_u|\mathbf{x}_u)}\Big] \\
& - D_{\mathrm{KL}}\big(p_{\mathrm{x}}(\mathbf{x}_u)\|p_{\mathrm{x|z}}(\mathbf{x}_u|\mathbf{z}_u)\big).
\end{aligned}
$$

$$(10)$$

which corresponds to Eq.(3).

As we have discussed in section 3.2, the direct optimization of the first term $\mathcal{L}_{\mathrm{direct}}$ might cause misalignment with the data likelihood maximization goal, since the observed data does not provide sufficient information about the retriever score distribution and the reranker is unaware of the noise patterns in the retriever score space. In contrast, our proposed solution solves this limitation by collaboratively learning the noise generator $f_\phi$ along with the reranker. The resulting objectives in Eq.(5) intentionally find a synthetic noisy score distribution $p_\phi$ that aligns with the real retriever score distribution and explores adversarial cases in which the reranker fails to accurately denoise into user feedback (details in Section 3.4).

## C LEARNING ALGORITHM

We present the overall learning process of our method in Algorithm 1.

---

**Algorithm 1:** Training procedures of DNR

---

**Input:** Hyperparameters $\lambda_c$, $\lambda_e$, and $\lambda_m$, initial reranker $\theta$ and noise generator $\phi$
**Output:** Optimized denoising reranker $q_\theta$ and the noise generator $f_\phi$.

1: **for** Epoch $i$ **do**
2:    **if** $i > \lambda_e$ **then**
3:       **for** Each sample $(\mathbf{u}, \mathbf{x}_u, \mathbf{z}_u)$ **do**
4:          Generate $\boldsymbol{\epsilon} \sim f_\phi^{\text{model}}$ for given $\mathbf{u}$ and $\mathbf{z}_u$ and form synthetic noisy scores $\mathbf{x}'_u$ with Eq (4)
5:          Optimize reranker $\theta$ based on $\mathcal{L}_\theta$;
6:          Optimize noise generator $\phi$ based on $\mathcal{L}_{\text{adv}}$ and $\mathcal{L}_{\text{x}}$;
7:       **end for**
8:    **else**
9:       **for** Each sample $(\mathbf{u}, \mathbf{x}_u, \mathbf{z}_u)$ **do**
10:         Sample $\boldsymbol{\epsilon} \sim f_\phi^{\text{heuristic}}$ for given $\mathbf{u}$ and $\mathbf{z}_u$ and form synthetic noisy scores $\mathbf{x}'_u$ with Eq (4)
11:         Optimize reranker $\theta$ based on $\mathcal{L}_\theta$;
12:       **end for**
13:    **end if**
14: **end for**

---

## D   DETAILED SETTINGS FOR OFFLINE EXPERIMENTS

### D.1   DATASETS

The ML-1M dataset is a public benchmark in the field of recommender systems, comprising 1 million ratings from 6,040 users across more than 3,900 movies. The Kuaivideo dataset is sourced from Kuaishou, a popular short-video app with over 300 million active users daily. The Amazon Books dataset is a comprehensive collection of product reviews specifically focused on books available on Amazon. For all datasets, we initially process by eliminating users and items with fewer than 20 interactions, to avoid the cold-start issue. For the retriever stage, we split the data into train and test subsets at a ratio of 8:2, where each sample is composed of interaction history and an item for prediction. For the reranker stage, we sort the interactions in chronological order, using the last six interactions as the item list revealed to users after reranking. We present the statistics of the processed datasets for the retriever and reranker stages in Table 5 and Table 6, respectively.

Table 5: Dataset Statistics for Retriever Stage

| Dataset | # Users | # Items | # Actions |
|---|---|---|---|
| ML-1M | 6,020 | 3,043 | 995,154 |
| Kuaivideo | 89,310 | 10,395 | 3,270,132 |
| Amazon-Toys | 35,732 | 38,121 | 1,960,674 |

Table 6: Dataset Statistics for Reranker Stage

| Dataset | # Users | # Items | # Sequences |
|---|---|---|---|
| ML-1M | 6,022 | 3,043 | 161,646 |
| Kuaivideo | 89,416 | 10,395 | 513,010 |
| Amazon-Toys | 35,736 | 38,121 | 311,386 |

### D.2   HYPERPARAMETERS SETTING

Here we present the optimal hyperparameters' settings in Table 7.

### D.3   BASELINES

We detail the compared baselines of our main experiments in the following, including traditional recommenders, list-refinement methods, generator-evaluator methods, and diffusion-based methods. **Traditional Recommenders:** predict the scores of candidate items and rank them accordingly.

- SASRec Kang & McAuley (2018) proposes a self-attention based sequential recommendation model (SASRec) that captures long-term semantics and short-term dynamics using an attention mechanism.
- Caser Tang & Wang (2018) introduces a convolutional sequence embedding recommendation model that captures sequential patterns using convolutional filters applied to latent space embeddings.

Table 7: Hyperparameter setting for DNR on different datasets

| Datasets | ML-1M | Kuaivideo | Book |
|---|---|---|---|
| $\lambda_c$ | 0.4 | 0.3 | 0.5 |
| $\lambda_m$ | 0.4 | 0.6 | 0.4 |
| $\lambda_e$ | 40 | 80 | 100 |
| weight_decay | 0.001 | 0.001 | 0 |
| dropout | 0.3 | 0.3 | 0.5 |
| batch_size | 2048 | 2048 | 2048 |
| emb_dim | 128 | 128 | 128 |

- GRU4Rec Hidasi et al. (2016) apply recurrent neural networks (RNNs) to session-based recommendations, addressing the challenge of short user sessions without long-term profiles.

- MiDNN Zhuang et al. (2018) proposes a global optimization framework for e-commerce search ranking that considers mutual influences between items, using extended features and sequence generation to optimize ranking and maximize overall utility.

**List-Refinement Methods:** encode the item lists from the previous stage and refine the rankings based on this information.

- DLCM Ai et al. (2018) leverages local ranking context from top-retrieved documents to refine initial ranking lists. The model uses a recurrent neural network to capture document interactions and an attention-based loss function, significantly improving performance over traditional learning-to-rank methods.

- SetRank Pang et al. (2020) introduces a neural learning-to-rank model that learns a permutation-invariant ranking function directly on document sets. Utilizing multi-head self-attention blocks, SetRank captures cross-document interactions and achieves robust performance across varying input sizes.

- PRM Pei et al. (2019) addresses personalized re-ranking in recommendation systems by incorporating user-specific preferences into the re-ranking process. It considers user behavior and candidate list, enhancing both relevance and personalization in the final recommendations.

- MIR Xi et al. (2022) proposes to capture complex interactions between user actions and candidate list features across multiple levels, improving the accuracy and relevance of recommendations through a hierarchical reranking approach.

**Generator-Evaluator Methods:** generate multiple ranked item lists and evaluate them to select the best one for users.

- EGRerank Huzhang et al. (2023) proposes an evaluator-generator framework for learning-to-rank (LTR) in e-commerce applications. The framework includes an evaluator that evaluates recommendations involving context, a generator that maximizes the evaluator score through reinforcement learning, and a discriminator that ensures the generalization of the evaluator.

- Pier Shi et al. (2023) follows a two-stage architecture with a Fine-grained Permutation Selection Module (FPSM) and an Omnidirectional Context-aware Prediction Module (OCPM). FPSM selects top-K candidate permutations based on user interest using SimHash, while OCPM evaluates these permutations with an omnidirectional attention mechanism.

- NAR4Rec Ren et al. (2024) explores the use of non-autoregressive generative models for re-ranking in recommendation systems, addressing challenges such as sparse training samples and dynamic candidates by introducing a matching model, unlikelihood training to distinguish feasible sequences, and contrastive decoding to capture item dependencies.

**Diffusion-Based Methods:** utilize diffusion models to generate items or lists for recommendation.

- DiffuRec Li et al. (2024) incorporates a diffusion process to inject noise into target item embeddings and a reverse process to reconstruct the target item representation.

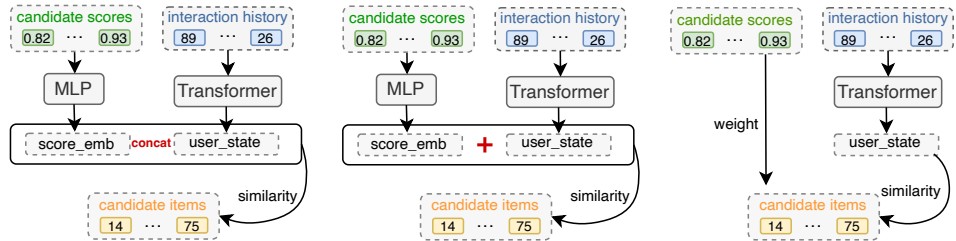

Figure 4: Different ways to leverage retriever scores.

- DCDR Lin et al. (2024) introduces a new framework that leverages diffusion models for the reranking stage in recommendation systems. DCDR extends traditional diffusion models with a discrete forward process and a conditional reverse process to generate high-quality item sequences.

### D.4    IMPLEMENTATION OF DIVERSE STRATEGIES FOR LEVERAGING RETRIEVER SCORES

Here, we present the detailed implementation of different ways to leverage retriever scores in Figure D.4. The left one refers to "c score", which leverages a concatenate operation to combine the score features with encoded user states from users' interaction history. The moderate one represents "+ score", which incorporates the score feature with user states by a simple plus operation. The right one is the "w score", which adopts the initial candidate score as the weight of candidate items.

## E    COMPLEMENTARY EXPERIMENTAL ANALYSIS

### E.1    EXPERIMENTS OF RETRIEVER STAGE

**Detailed Experimental Settings for Retriever Stage**

We implement the retriever stage with the MF method Koren et al. (2009), which aims to improve the click rate performance. The dimension of the item embedding is 128, and the learning rate is 0.001. We randomly select negative items to train the MF method, enhancing the retriever's performance despite the item sparsity.

**Experimental results of Retriever Stage**

Here, we present the detailed results of the retriever stage in Table 8.

Table 8: Performance of the retriever stage, which adopts the matrix factorization (MF) model to rate and select the top-50 items from full item set.

| Datasets | ML-1M | Kuaivideo | Book |
|---|---|---|---|
| AUC | 79.20% | 89.62% | 88.82% |

### E.2    VISUALIZATION OF LEARNED NOISES

To visualize the generated noise distribution, we compare the density curve of the generated noise from different noise generators, including Gaussian noise, Beta noise, and learning-based noise. As demonstrated in Figure 5, the distribution of learned noise is more aligned with the distribution of true noise between retriever scores and user feedback. This means that the resulting posterior with model-based noise sampler better reduces the divergence term $\delta_x$ in Eq.(5), and potentially improve the alignment between our proposed solution with the three objectives and the data log-likelihood maximization goal.

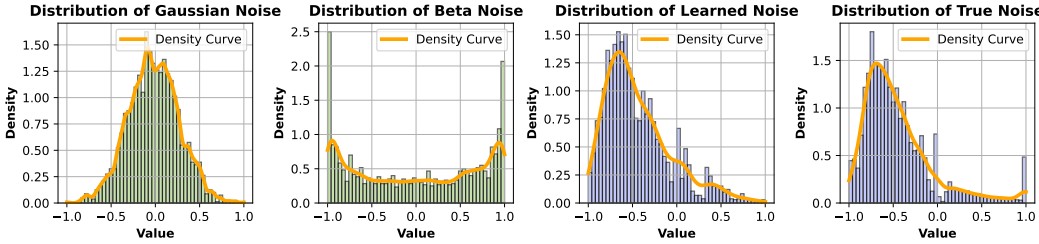

Figure 5: The visualization of generated noise from different variants.

### E.3 PERFORMANCE ON DIFFERENT SETTINGS OF BETA-DISTRIBUTION

The U-shaped Beta distribution in our paper is motivated by the binary nature of the user feedback scores and the uncertain initial guess, and it is already empirically outperforming the Gaussian distribution in our experiments. Since the heuristically designed beta distribution is not guaranteed to be optimal — unlike the model-based noise generator, whose learned noise distribution is demonstrated to be closer to the real noise distribution.

To explore different Beta distributions, we conducted additional experiments with DNR that used a beta distribution Beta(2, 5), *i.e.,* DNR-BL, which is more aligned with the true noise distribution as well as our model-based generator (as shown in Figure 5), before switching to the model-based generator in epoch $\lambda_e$. Empirically, the results are presented in Table 9.

Table 9: Experiments with the beta distribution replicate the learned distribution.

| Methods | ML-1M | | Kuaivideo | | Book | |
|---|---|---|---|---|---|---|
| | NDCG@6 | MAP@6 | NDCG@6 | MAP@6 | NDCG@6 | MAP@6 |
| DNR-B | 77.12 | 67.30 | 70.15 | 58.12 | 83.53 | 75.54 |
| DNR-BL | 77.43 | 67.49 | 71.10 | 59.61 | 83.68 | 75.79 |

We can observe that using this more "aligned" beta distribution (*i.e.,* , DNR-BL) before epoch $\lambda_e$ offers a slight improvement compared to using the U-shaped Beta (*i.e.,* DNR-B) before $\lambda_e$. This reinforces our main claim: adopting a model-based noise generator to learn and regularize noise distribution, aligning it more closely with the ground-truth noise distribution.

### E.4 DETAILED RESULTS OF SENSITIVITY ANALYSIS

Here, we present the detailed results of the sensitivity analysis for different datasets in Figure 6, Figure 7, and Figure 8.

### E.5 EXPERIMENTS ON LARGER RERANKER BACKBONES

DNR is a backbone-agnostic framework for reranking tasks, utilizing and denoising the noisy scores from the previous retrieval stage through adversarial learning. Recently, LLMs have become a powerful tool for reranking tasks Mao et al. (2025b). However, text-based LLM4Rerank is not comparable in our setting offline or online, due to the lack of rich textual information (such as detailed item descriptions) that LLM-based rerankers fundamentally rely on. To evaluate the effectiveness of DNR in large models, we scaled up our PRM baseline by significantly increasing the number of its Transformer layers, serving as a "larger model" for comparison. We then tested our DNR framework, using this scaled-up PRM as its reranker backbone. The results are presented in the table 10.

We can observe that scaling up the backbone may improve the reranking result, comparing the larger PRM with the original PRM, or comparing the larger DNR with the original DNR. Additionally, our original DNR outperforms the scaled-up "larger model" baseline (larger PRM); The DNR framework may also generalize well to a larger model.

Table 10: Experiments on larger models (expand transformers of PRM to 10 layers) with ML-1M dataset.

| Methods | ML-1M | | | | |
|---|---|---|---|---|---|
| | HR@6 | NDCG@6 | MAP@6 | F1@6 | AUC |
| original PRM | 60.09 | 72.85 | 62.21 | 65.51 | 88.20 |
| original PRM+DNR | 64.89 | 77.67 | 68.00 | 70.71 | 92.75 |
| larger PRM | 62.95 | 76.09 | 66.72 | 68.61 | 88.68 |
| larger PRM+DNR | 65.87 | 78.72 | 69.33 | 71.95 | 89.19 |

### E.6 ONLINE A/B TEST

To better investigate the effectiveness of our DNR in an online environment, we conduct an A/B test on an industrial platform with video feeds to 100+ millions of users daily.

**Detailed Experimental Settings for Online Test** The multi-stage pipeline of the online system consists of three stages: retrieval, ranking, and reranking. In the final reranking stage, the top-6 items for user exposure are selected from a candidate set of 50 items generated by the ranking stage. During the ranking process, a scoring model computes multi-dimensional features for each item. Among these, we utilize PCTR(predicted probability that the video will be watched for more than 3 seconds) as the primary retrieval score $\mathbf{x}_u$, which corresponds to the "realshow" metric in the online system. Note that the final reranking stage considers the combination of the retrieval stage and the middle ranking stage as a holistic previous stage, so the output PCTR scores from the ranking stage are considered as "retrieval score" in this final stage. DNR is applied to generate a new set of scores for 50 selected items and the top-6 items are chosen as the final exposure.

**Experiment Results** For our online experiments, the total traffic is randomly divided, with 12.5% allocated to the DNR method and 12.5% to the baseline method. Each experiment is conducted for at least 7 days to ensure the reliability of the results. The DNR method demonstrates a significant improvement in realshow(, *i.e.,* the cumulative number of videos watched by users), increasing it by 1.089%. These results indicate that the DNR method can estimate PCTR more accurately, which is strongly correlated with realshow. Additionally, the industrial recommendation environment involves various other metrics in addition to the realshow metric, including but not limited to app usage time, watch time, share-rate, like-rate, and comment-rate. We provide the corresponding results in Table 11. We can see that, in our scenario, watch-time and share-rate are slightly negatively impacted, while like-rate and comment-rate are slightly positively impacted, indicating a potential trade-off between metrics. Notably, while the watch time exhibited a slight decrease, we additionally observed that the long-term DAU slightly increased by 0.01% [-0.012%,+0.012%], which is nearly reaching a statistically significant improvement. This potentially suggests that the slight reduction in viewing duration is likely attributable to users exploring a broader range of clicked videos, while more significant long-term indicators remain positive. Beyond these, the impact on all other metrics is not statistically significant, which validates the effectiveness of our method. It is worth noting that achieving performance gains across all online metrics is inherently a multi-objective challenge—an area that remains a key focus for future work. Additionally, we believe applying DNR to other xTR signals (e.g., watch time and share-rate) holds significant potential for further improvements.

### E.7 EXPERIMENTS ON DIVERSE RETRIEVERS

To better illustrate the generalizability of DNR across different retrievers, we implemented the dual-tower (DT) model as a stronger retriever. The comparison results in the retrieval stage are summarized in Table 12, and the comparison results in the reranking stage are summarized in Table 13. We can see that DT+DNR outperforms DT+PRM, validating the generalizability of our method and the sustained advantage of the denoising framework.

Theoretically, there will always be an empirical error that upper bounds the overall recommender system, which aims to minimize the "optimizable error", and there will be no room for the reranker to "denoise the scores" if the retriever itself achieve this bound. Yet, this theoretical bound is still

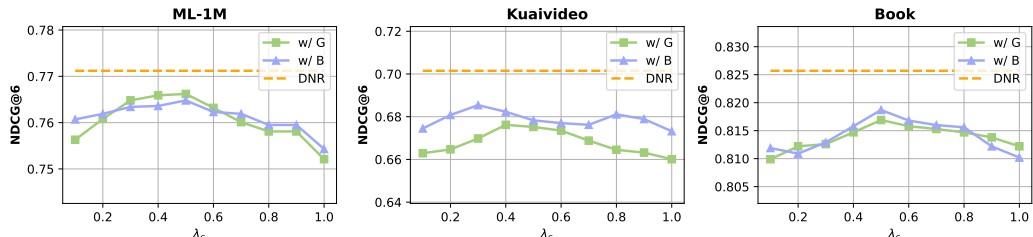

Figure 6: Sensitivity of DNR to the hyperparameter $\lambda_c$ on different datasets, which controls the degree of noise injection.

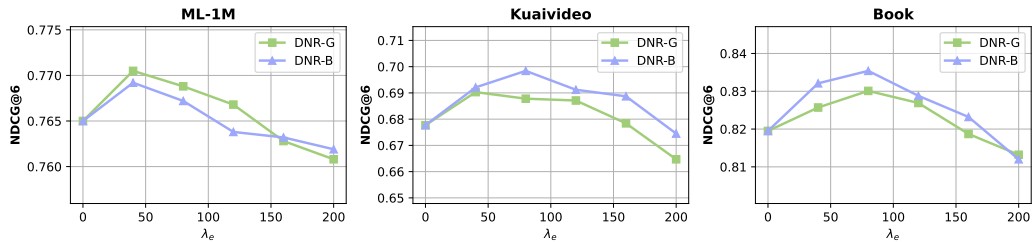

Figure 7: Sensitivity of DNR to the hyperparameter $\lambda_e$ on different datasets, which decides the epochs where the adversarial learning starts.

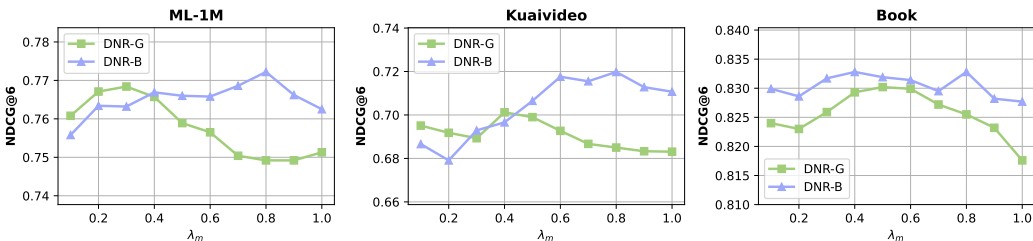

Figure 8: Sensitivity of DNR to the hyperparameter $\lambda_m$ on different datasets, which controls the magnitude of the augmentation.

| metric | performance boost | statistically significant ($p < 0.05$) |
|---|---|---|
| overall app-time | -0.011% | no |
| realshow | +1.089% | yes |
| watch-time | -0.091% | no |
| share-rate | -0.758% | no |
| like-rate | +0.609% | no |
| comment-rate | +0.574% | no |
| save-rate | -0.690% | no |
| profile-click-rate | +0.129% | no |
| Long term DAU | +0.01 % | yes |

Table 11: Extended online A/B test results.

impractical for a single retriever in most industrial platforms, due to various reasons including (but not limited to) the computational bottleneck, multi-task trade-off, and filtering on large-scale dynamic candidate pool. A more advanced retriever means a smaller remaining error that can be optimized, which indicates that the reranker will solve a simpler denoising problem and simpler reranker design. In this case, a diminishing marginal gain only indicates a diminishing computational requirement in the reranking stage. In general, the advantage of the denoising formulation is maintained regardless of the retriever employed.

Table 12: The AUC performance of the different retrievers, including the matrix factorization (MF) model and dual-tower model.

| | ML-1M | Kuaivideo | Book |
|---|---|---|---|
| MF | 79.20% | 89.62% | 88.82% |
| DT | 79.86% | 90.38% | 90.0% |

Table 13: The performance comparison of rerankers on different retrievers.

| Methods | ML-1M | | Kuaivideo | | Book | |
|---|---|---|---|---|---|---|
| | NDCG@6 | MAP@6 | NDCG@6 | MAP@6 | NDCG@6 | MAP@6 |
| MF+PRM | 72.85 | 62.21 | 55.93 | 42.97 | 76.88 | 66.44 |
| MF+PIER | 75.99 | 65.98 | 65.11 | 52.55 | 80.22 | 71.62 |
| MF+DNR | 77.67 | 68.00 | 70.15 | 58.12 | 83.53 | 75.54 |
| DT+PRM | 76.68 | 67.37 | 65.77 | 53.41 | 81.18 | 72.72 |
| DT+PIER | 76.36 | 66.35 | 68.34 | 56.26 | 81.55 | 72.79 |
| DT+DNR | 77.96 | 68.78 | 70.96 | 58.87 | 84.42 | 77.11 |

### E.8 EXPERIMENTS ON DNR'S DIVERSITY

Since biases in the retriever's scores may not align with ground-truth user preferences, they will be "denoised" if they constitute noise in the score signal. Unlike alternative methods that leverage retriever scores directly, DNR's primary advantage lies in denoising unreliable scores enabled by its noise-aware training paradigm, as evidenced by Table 2. For example, a specific user may not like a certain video even if it is popular. When the noise-aware DNR finds this noise and corrects it, the resulting prediction becomes more accurate, more personalized, and more diverse.

We conduct additional experiments on the diversity of the final recommendations by comparing the Coverage@1 and coverage@6 metrics of the rerankers' outputs. As shown in Table 14, the experimental results demonstrate that DNR's denoising capability enables it to mitigate incorrect biases in retriever scores, thereby achieving greater diversity compared to baseline rerankers (*e.g.,* PRM and PIER).

Table 14: The diversity performance comparison of DNR and different rerankers.

| Methods | ML-1M | | Kuaivideo | | Book | |
|---|---|---|---|---|---|---|
| | C@1 | C@6 | NC@1 | C@6 | C@1 | C@6 |
| PRM | 0.5366 | 0.9080 | 0.7609 | 0.9648 | 0.4364 | 0.8584 |
| Pier | 0.5429 | 0.9073 | 0.7554 | 0.9571 | 0.4238 | 0.8663 |
| DNR | 0.5626 | 0.9109 | 0.7923 | 0.9702 | 0.4786 | 0.8721 |

## E.9 COMPUTATIONAL ANALYSIS

Theoretically, our method does not change the computational cost during inference since we can adopt the same reranking model structure (*e.g.,* PRM Pei et al. (2019)). In the training phase, we include an additional noise generator that may require O(1) computational overhead, whose cost depends on the generator's structure. Empirically, the generator does not have to be sophisticated, and the computational overhead can be trivial compared to the reranker's training.

To better demonstrate computational cost, we compare the per-epoch training time of DNR against several baseline rerankers, including PRM Pei et al. (2019), Pier Shi et al. (2023), DCDR Lin et al. (2024), and a "w/ score" variant—where retriever scores are directly used as input to the reranker. In table 15, we present the time cost comparison for different reranker baselines in each training epoch. The variant "w/ score" denotes methods incorporating retriever scores as input features, regardless of the specific integration method (*e.g.,* concatenation, plus, or weighted combination). Our proposed DNR demonstrates competitive training efficiency, with time costs comparable to both PRM and w/ score baselines across all datasets (ML-1M, KuaiVideo, and Book). This validates the computational efficiency of our approach while maintaining competitive performance.

Table 15: Comparison of time costs.

| Datasets | ML-1M | Kuaivideo | Book |
|---|---|---|---|
| PRM | 01m01s | 02m21s | 01m42s |
| Pier | 01m54s | 04m11s | 02m36s |
| DCDR | 02m27s | 04m58s | 03m02s |
| w/ score | 01m03s | 02m49s | 01m46s |
| DNR | 01m06s | 03m12s | 01m51s |

## F LIMITATION AND DISCUSSION

### F.1 LIMITATIONS

**Computational Cost:** Compared to a direct optimization framework with Eq.(2) that only learns $\theta$, our framework requires extra efforts to train a posterior model $\phi$ (if using learnable posterior) with three additional loss terms to back-propagate. This might multiply the training cost for several folds. Fortunately, $\phi$ is not involved in inference, so the inference cost stays the same.

### F.2 FUTURE DIRECTIONS

**Multi-task Scenarios:** When a user can engage with an item with multiple behaviors (*e.g.,* like and share a news story), DNR may adapt to this scenario by simply formulating a multi-dimensional response space for $\mathbf{z}_u$ and $\mathbf{x}_u$. Yet, different behavior signals may typically follow a heterogeneous distribution where a uniform modeling technique might become sub-optimal. Additionally, the problem may become even more sophisticated when there exist multiple retrievers that estimate the same metric, especially in industrial settings. In this case, one may have to come up with a more articulated integration method to clarify the connections between the user feedback and different retriever scores. Thus, we believe it is worth further investigating how the proposed framework may accommodate the multi-task solutions.

**Joint Optimisation of Multi-stages:** Since our "retriever-aware reranker" is a complementary piece to the "reranker-aware retriever" work, we believe optimizing the two components end-to-end jointly is a very promising direction for future research. The potential lies in tackling the core multi-stage challenges: effectively aligning the divergent training objectives, reliably back-propagating non-uniform feedback signals across the stages, and allowing flexible model switch for any stage.

**Multi-stage vs. End-to-End:** Recent advances in end-to-end recommenders have witnessed significant advances, and evidence Deng et al. (2025); Kong et al. (2025) has shown that end-to-end recommenders are potentially more effective since they no longer need to worry about the consistency challenge in the multi-stage framework. Yet, there is still no evidence of a general superiority comparing these two paradigms, and there is still no optimal solution for a fully consistent multi-stage recommender.

## G  USE OF LLMS

Large language models (LLMs) were used only to aid writing polish—including refining sentence phrasing, logical flow, and prose clarity—without altering original meanings or technical details. LLMs did not participate in core research tasks (*e.g.,* experiment design, data processing, model training, result analysis, or drafting key technical content).

