# OpenReview forum: "Denoising Neural Reranker for Recommender Systems"
_ICLR.cc/2026/Conference — ICLR 2026 Poster_

### Official Review · Reviewer_6Pyu · 2025-10-25

**Soundness:** 2
**Presentation:** 1
**Contribution:** 2
**Rating:** 4
**Confidence:** 4

**Summary:**

This paper introduces a novel “recall-score denoising” perspective and designs an adversarial training framework, DNR, that enables a reranking model to exploit recall scores more effectively, leading to substantial improvements in recommendation quality. The effectiveness and universality of the proposed method are verified through both offline and online experiments. In addition, the experimental section formulates six research questions and sequentially validates the overall efficacy of the approach.

**Strengths:**

1. Focusing on multi-stage recommender systems, the paper argues that prior work overwhelmingly concentrates on “reranking-aware retrievers” while overlooking “retriever-aware rerankers”. Although retriever scores are information-rich, they are simultaneously noisy; reranking can therefore be viewed as a denoising procedure. Naïvely exploiting these scores—e.g., simply treating them as additional features: suffers from theoretical limitations and yields poor alignment with actual user feedback. To this end, we propose DNR (Denoising Neural Reranker), which couples a denoising reranker qθ with a learnable noise-generation module fϕ.

2. The user-feedback alignment objective is decomposed into three losses: (i) Lz, the denoising loss, trains the reranker to discriminate observed from synthetic scores; (ii) Ladv, the adversarial loss, encourages the generator to produce hard-to-denoise samples; and (iii) Lx, a KL-regularizer, aligns the synthetic score distribution with the empirical retriever distribution.

3. Experiments on ML-1M, Kuaivideo and Amazon-Books compare DNR against both traditional recommenders (e.g., SASRec) and specialized reranking baselines (PRM, PIER). Six research questions (RQ1–RQ6) systematically demonstrate DNR’s superiority, including its generalization capacity and the impact of different noise generators. An industrial Online A/B test further confirms its practical utility.

**Weaknesses:**

1. Several figures are of sub-optimal quality and remain to be refined (e.g., Fig. 1(b) and Fig. 4); likewise, some textual descriptions and typesetting details, such as the presentation of RQ1–RQ6 in Section 4 EXPERIMENTS: call for improvement. Overly long sentences (e.g., the multi-stage architecture description in the introduction) impair readability.

2. The proposed DNR framework is thoroughly compared with a variety of state-of-the-art reranking baselines (PIER, NAR4Rec, DCDR, etc.). The experimental protocol is comprehensive and the results are convincing. Nevertheless, a highly related recent work: “Comprehensive List Generation for Multi-Generator Reranking”(has neither been cited nor included in the comparison). Like DNR, it focuses on enhancing the reranking stage’s capacity to model candidate lists. Incorporating this baseline would offer a more complete assessment of DNR’s relative merits and would strengthen the paper’s timeliness within the rapidly evolving reranking literature.

3. In the online A/B test, RealShow improves markedly, whereas the remaining metrics are either negative or neutral; however, no explanation is provided. RealShow denotes the cumulative number of users who watched videos, yet overall app-time and watch-time decrease. This pattern suggests that the optimized model may surface content that users are willing to click but do not actually prefer, implying that the deployed model’s performance could be inferior to its predecessor. A detailed analysis is warranted.

4. When introducing the noise generator in Section 3.3, the authors borrow directly from diffusion models (Ho et al., 2020) and VAEs (Kingma & Welling, 2014), but they do not justify why the Beta distribution is a “natural” prior or why a model-based perturbation scheme should be preferred over heuristic alternatives: empirical verification alone is supplied.

**Questions:**

1. The legend in Fig. 1(b) obscures the main pattern; arranging the four sub-figures in a 2×2 grid would alleviate the overlap.
2. The font color in Fig. 4 is sub-optimal and markedly reduces legibility; a higher-contrast palette should be adopted.
3. Several passages lack clarity and fluency. For example, “Recommender systems aim to personalize a list of items that maximizes user-engagement signals, such as purchases in e-commerce, video plays in video platforms, and link sharing in social networks. To achieve a feasible solution with large-scale candidate pools and user requests, industrial solutions … typically employ a multi-stage architecture comprising a retriever and a reranker.”—should be split into two or three shorter sentences to enhance readability.
4. Equation (5) decomposes −log p(z_u) into L_z + L_adv + L_x + δ_x, where δ_x = −D_KL(p_ϕ‖p_{x|z}) ≤ 0 is assumed to be “small” and is indirectly minimized through other losses. The manuscript, however, offers neither a convergence guarantee nor numerical evidence to justify this assumption; the motivation for treating δ_x as negligible remains unexplained.

---

> ### Author Response · Authors · 2025-11-21
> **Response to Reviewer 6Pyu --- Part 1/3**
>
> We sincerely appreciate your constructive and detailed comments on our work.  For the weaknesses and questions, we have detailed our responses point-by-point to your comments. We sincerely hope that our additional response has adequately addressed your concerns.
>
> ***1) Presentation Quality:***
>
> We will address these presentation issues as follows in our revision, but please note that they do not affect our claims about the proposed DNR method.
> - Fig. 1(b): After incorporating an additional motivational sub-figure illustrating distribution divergence (as suggested by Reviewer Xkya), the revised Figure 1 will rearrange the five sub-figures into a 2×3 grid. And as you suggested, we will also prevent the legend from obscuring key patterns and enhance the overall clarity of the figure.
> - Fig. 4: We will replace the color palette with a high-contrast, accessibility-friendly one to ensure immediate legibility in our revision.
> - Typesetting: We will reformat the presentation of RQ1–RQ6 to be cleaner and a more direct listing manner in our revision, rather than a dense paragraph.
> - Long sentences: We will segment the overly long sentences into shorter ones to increase readability.
>
> ***2) More baselines:***
>
> We conduct additional experiments on MG-E(CLIG) according to the mentioned paper ("Comprehensive List Generation for Multi-Generator Reranking"), and the results are presented in Table R4. As shown, DNR outperforms MG-E and achieves the state-of-the-art performance, which may be attributed to its ability to leverage retriever scores and denoise them. We will add these results and analysis in our revision.
>
>
> **Table R4:** The performance comparison of MG-E and DNR.
> | Methods | ML-1M|  | Kuaivideo  |   | Book   |  |
> | :---:   | :-----: | :------: | :------: | :------: | :------: | :------: |
> |        | NDCG@6| MAP@6 | NDCG@6| MAP@6 | NDCG@6| MAP@6|
> | PIER | 75.99| 65.98|65.11 |52.55|80.75 |72.40|
> | MG-E |76.18|66.52|66.19| 53.75|79.81|71.01|
> | DNR | 77.67 | 68.00|70.15|58.12|83.53|75.54|

---

> ### Author Response · Authors · 2025-11-21
> **Response to Reviewer 6Pyu --- Part 2/3**
>
> ***3) Online metric trade-off:***
>
> **Clarification about metrics:** In our system, all of the following metrics are considered as "short-term" engagement metrics that contribute to the core metric app-time: watch-time, share, like, comment, save, and profile clicks. Typically, the watch-time is not necessarily a long-term metric but rather an engagement metric, since it is directly connected to the user's watching behavior on each video. In contrast, the realshow metric describes the number of effective viewing behaviors (but not the cumulative number of users), which determines the support of the "engagement" as well as the "app-time", which is usually independently optimized.
>
> **Statistical significance:**
> As detailed in Appendix E.6 (Table 11), the slight decline you noted in watch-time (-0.091%), app-time (-0.011%), and share-rate (-0.758%) **was not statistically significant** (p > 0.05). Empirically, the watch-time has an empirically significant region beyond [-0.16%,+0.16%], the app-time's region beyond [-0.09%, +0.09%], and all other engagement metrics have empirical regions beyond [-1.3%,+1.3%]. In contrast, **our primary optimization was conducted on the binary PCTR** (corresponding to realshow metric), and achieved a +1.089% gain that was statistically significant (empirically significant region is beyond [-0.5%,+0.5%]), demonstrating the effectiveness of DNR.
>
> **Trade-off between metrics:**As we have described in Appendix F.2, achieving performance gains across all online metrics is inherently a **multi-objective task**, which remains a key area for future work. In Table 11, some engagement metrics like watch-time and share-rate slightly decrease, but other engagement metrics like like-rate and comment-rate increase, which reveals a trade-off behavior among metrics other than the realshow, instead of a simultaneous drop for all metrics. While we have only conducted experiments on the realshow metric, we believe that **applying DNR to other xTR signals** also holds potential for further improvements.
>
> **Long-term performance:**
> To better address the reviewer's concern, we additionally observed that the **long-term DAU slightly increased by 0.01% [-0.012%,+0.012%]**, which is nearly reaching a statistically significant improvement. This potentially suggests that the slight reduction in viewing duration is likely attributable to users exploring a broader range of clicked videos, while more significant long-term indicators remain positive. In other words, the users may have found their time more effectively used in the system since more valuable information is consumed in a shorter time range. This may also address the reviewer's concern about "clickbait" videos. We will supplement the long-term DAU in Table 11 of Appendix E.6.
>
> **Clickbait concerns:**
> To further address the concern about the clickbait issue, we additionally observed the "hate-rate" metric, since an increase in clickbait videos would empirically cause an increase in this metric. In our online experiment, we find that this **hate-rate is actually decreased by -0.178% [-0.19%,+0.19%]** in the A/B test (close to statistical significance), which indicates that DNR may help reduce such unwanted videos and improve the experiences of users. This may also contribute to the slightly increased long-term DAU.
>
> ***4) Connection to diffusion and VAEs:***
>
> As we have described in Appendix B.1, our method is similar to a one-step diffusion in the sense of problem formulation, but a key difference is that the real retriever scores are not necessarily sampled by a manually defined distribution, but an existing retriever prior. Additionally, the variational inference technique used in VAEs inspired our derivation of the final solution, but it is different in the reconstruction direction, where z, instead of x, is reconstructed since z is now observable. The resulting objective decomposition is thus different from the original ELBO derivation.
>
> ***5) Why Beta distribution:***
>
> As we have mentioned in the problem formulation, we are focusing on binary user feedback that is widely used in recommender systems. Following probabilistic theories, the conjugate prior and the posterior for **binary variables** (i.e., Bernoulli or binomial distribution) are both beta distributions. **While the Gaussian distribution is widely used in vision tasks (where the representation space is nearly a continuous space), it is not necessarily the best choice in recommender systems**.
> In our empirical results, we find the Beta distribution works better than the Gaussian distribution in most cases. Regardless of the choice of the heuristic, the overall denoising framework and the DNR solution are supported by our empirical and theoretical analysis in this paper.

---

> ### Author Response · Authors · 2025-11-21
> **Response to Reviewer 6Pyu --- Part 3/3**
>
> ***6) Theoretical motivation for model-based perturbation***
>
> As we have described in section 3.3, it is reasonable to argue that any heuristic design could be suboptimal when designing the posterior. The corresponding theoretical motivation could be explained through the following perspectives:
> - Personalisation: to guarantee statistical accuracy, the heuristic design usually applies the same hyperparameter setting across all users, while the model-based posterior can generate personalized noises, which potentially improves the overall robustness.
> - Direct optimization: note that the heuristic solution does not optimize $L_\text{x}$ and $L_\text{adv}$, and one can only choose different design choices to reduce these losses, which takes excessive human efforts. In contrast, the model-based posterior directly optimizes these two objectives by adversarial learning and noise regularization, which is a more adaptive design that better exploits the data.
> - Generalization: Mechanically, the heuristic-based solution can be regarded as a special case of the model-based solution, where the learned model is exactly the heuristic. As a result, the model-based solution is fundamentally more expressive in practice. Since the precise noise distribution is uncertain, the model-based generator provides the necessary mechanism to learn this distribution.
>
>
> We will update Section 3.3 to include this clear clarification for both the Beta distribution (due to the binary [0, 1] bounded nature of the scores) and the model-based generator (to learn complex systemic noise adaptively, not just simple noise).
>
> ***7) Justification for Eq.5:***
>
> The primary theoretical reason for "expect to achieve a small $\delta\_\text{x}$" is that the regularization term in $\mathcal{L}\_\text{x}$ will strictly bound $\delta\_\text{x}$, as:
>
> $$
> \begin{aligned}
> \mathcal{L}\_\text{x} &= D\_\text{KL} \big( p\_\phi(\mathbf{x}|\mathbf{z}\_u) || p\_\text{x}(\mathbf{x}) \big) \\\\&= \mathbb{E}\_{p_\phi}\big[\log \frac{p\_\phi}{p\_\text{x}}\big] \\ = \mathbb{E}\_{p_\phi}\big[\log \frac{p\_\phi}{p\_\text{x|z}}\big] + \mathbb{E}\_{p\_\phi}\big[\log \frac{p\_\text{x|z}}{p\_\text{x}}\big] \\\\&= -\delta_\text{x} - H(p\_\phi, p\_\text{x|z}) + H(p\_\phi, p\_\text{x})\\\\ &\Rightarrow -\delta\_\text{x}  = \mathcal{L}\_\text{x} + H(p\_\phi, p\_\text{x|z}) - H(p\_\phi, p\_\text{x})
> \end{aligned}
> $$
>
> In other words, the KL divergence inside $\delta_\text{x}$ will be implicitly minimized through $\mathcal{L}_\text{x}$.
>
> Additionally, the cross entropy term $H(p_\phi, p_\text{x|z})$ also represents how the posterior distribution is different from the ground-truth posterior, which is implicitly optimized through $\mathcal{L}\_\text{x}$. The remaining term $- H(p_\phi, p_\text{x})$ estimates the alignment between the posterior and the ground-truth prior. Note that the adversarial loss $\mathcal{L}\_\text{adv}$ aims to find adversarial samples that the reranker finds hard to optimize, which also tends to explore special $\mathbf{x}$s that are different from the prior distribution. In this sense, we can expect the negative cross entropy $- H(p\_\phi, p\_\text{x})$ to have a lower value.
>
> On the other hand, considering the relationship between the adversarial loss and $\delta_\text{x}$, we have the following derivation:
>
> $\mathcal{L}\_\text{adv} = \mathbb{E}\_{p\_\phi}\big[ \log \frac{q\_\theta}{p\_\text{z|x}} \big] = \mathbb{E}\_{p_\phi}\big[ \log \frac{q\_\theta}{p\_\text{z|x}} \frac{ p\_\text{x}}{p\_\text{z}} \frac{ p\_\text{z}}{p\_\text{x}} \big]\\
> =\mathbb{E}\_{p\_\phi}\big[ \log \frac{\frac{q\_\theta p\_\text{x}}{p\_\text{z}}}{p\_\text{x|z}} \big]$,
>
> where we can assume that $q\_\theta p\_\text{x} / p\_\text{z}$ is bounded by $p_\phi$ due to the augmented reconstruction loss $\mathcal{L}\_\theta$. This means that $\delta\_\text{x}$ is also implicitly bounded by $\mathcal{L}\_\text{adv}$.
>
>
> We sincerely hope that our additional response could address your concerns. If there are any remaining concerns, please let us know, and we will continue to actively address your comments and further improve our work.

---

> ### Author Response · Authors · 2025-11-26
> **Kindly Reminder**
>
> Dear Reviewer 6Pyu,
>
> We deeply appreciate your valuable feedback and the time you've taken to review our work, especially during this busy period.
>
> We are reaching out to kindly inquire about the current status of your review regarding our submission. We sincerely hope that our responses have adequately addressed your concerns. Furthermore, we are eager to address any additional queries you might have, which will enable us to enhance our work further.
>
> Once again, thank you for your guidance and support.
>
> Best, Authors

---

### Official Review · Reviewer_Xkya · 2025-10-29

**Soundness:** 3
**Presentation:** 4
**Contribution:** 3
**Rating:** 8
**Confidence:** 4

**Summary:**

This paper focuses on designing a retriever-aware reranker for multi-stage recommendation. Motivated by the empirical observation that the reranker may serve as a denoiser for the retriever, the authors propose an adversarial denoising framework DNR, which incorporates three components beyond the original reranking objective: (1) denoising loss, (2) adversarial noise generator loss, and (3) score distribution KL divergence term. Comprehensive experiments on three public datasets and one online A/B test on the industrial platform demonstrate the effectiveness of the proposed method.

**Strengths:**

- The proposed objective is theoretically well-motivated, which serves as an upper bound of the negative log-likelihood of the data.
- The experiments are comprehensive, including both offline comparisons with baselines, sufficient ablation studies, and an online A/B test, which validates the effectiveness of the proposed method.
- The writing and derivation is clear and easy to follow.

**Weaknesses:**

In general, this paper is well-written and supports its claims with sufficient experiments. No major weaknesses are found, while some questions are raised below.

**Questions:**

**Motivation.** This paper is motivated by the empirical observation that the reranker serves as a denoiser for the retriever. However, the explanation for this claim in Figure 1 is not very clear, as the T-SNE of the embeddings does not directly reflect the denoising effect. I suggest the authors show the distribution divergence between the retrieved/reranked scores and the ground-truth labels to directly support this claim.

**Comments on Experiments.** Some clarifications on the experimental results are needed:
- In Figure 3, the performance of DNR with different noise proportions $\lambda_c$ stays the same, and why?
- In Figures 3 and 6-7, the NDCG@20 performance is reported. Compared to NDCG@6 results in Table 2, is the results in these figures are actually NDCG@6? Moreover, the peak metrics of the three curves in Figure 3 are not the same, why?

**Compatibility with Reranker-aware Retriever Methods.** There are various works on designing reranker-aware retrievers, as mentioned in Section 2.1. Can the proposed DNR be combined with these methods to further improve the overall performance?

---

> ### Author Response · Authors · 2025-11-21
> **Response to Reviewer Xkya**
>
> We are deeply grateful for your positive review and "accept" recommendation. Thank you for recognizing our work as "theoretically well-motivated," "comprehensive," and "clear". We will gladly address your excellent questions to make the paper even better.
>
>
> ***Question 1: Clarity of Motivation (Figure 1)***
>
> To more vividly substantiate the claim regarding noise, we present a comparative analysis of the noise distributions between retriever and reranker scores. The density curves illustrate the discrepancies between predicted scores and ground-truth labels. **We have included it as a sub-figure of Figure 1 in our revision**. As evident from these distributions in the figure, the noise between the reranker and ground-truth labels is more tightly concentrated around 0. In contrast, the noise associated with the retriever and ground-truth is predominantly centered near -0.5, indicating a substantially higher magnitude of deviation. These findings demonstrate that the retriever’s scoring outputs contain non-negligible noise relative to both the reranker’s outputs and the ground-truth labels—thus affirming the reranker’s role as an effective denoising component for the retriever.
>
> ***Question 2: Clarification on Figures 3, 6, 7***
>
> We sincerely apologize for the typos and lack of clarity in these figures.
> - Typo Correction: We are sorry for this mistake. The y-axis in these sensitivity plots should be NDCG@6, not NDCG@20. We will update them in our revision and thank you so much.
> - Clarification for Figure 3
>   - "Flat line" (Fig 3 left): The hyperparameters $\lambda_c$ are tuned particularly on heuristic-based solution, representing the proportion of noise in the generated retriever scores. While for the Model-based noise generator, the noise is entirely determined by the model $\phi$, so one can **regard $\lambda_c=1$ after switching**. Before the switching, we use the optimal setting (in Table 7) for the heuristic-based noise generator. **The dashed "DNR" line represents the performance of our final, fully-trained framework (using the model-based solution)**. It is presented as a flat line merely for reference, which illustrates that the final model outperforms the heuristic generators ("w/ G", "w/ B") across all $\lambda_c$ values. We will revise our paper and incorporate this explanation to enhance clarity.
>   - "Different peaks": Figure 3 shows the sensitivity of three different hyperparameters. To keep a fair comparison, we fixed all other hyperparameters and tuned the target hyperparameter. For the left figure, we fix $\lambda_m=0.6$ and $\lambda_e=80$. For the middle figure, we fix $\lambda_e=40$. For the right figure, we fix $\lambda_m = 0.3$. **The discrepancies in the peak values are a result of the differing fixed hyperparameter configurations**. Note that DNR-G and DNR-B share the same parameter tuning space, but their choices of different distributions caused the differences in their peak settings.
> We will correct all errors and provide additional clarifications to render these points explicit and easily understandable.
>
> ***Question 3: Compatibility with Reranker-aware Retrievers***
>
> Intuitively, this could be a very promising and impactful direction since one can assume that any stage in the multi-stage recommender is solving an intermediate denoising problem. The resulting framework would consider the whole multi-stage system as a multi-step diffusion and may solve the inconsistency challenge with a theoretically sound solution. **We will note this as a key area for future work in Section F.2 in our revision**.
>
>
>
> Thank you again for your supportive and constructive review! If there are any remaining questions, please let us know, and we will continue to actively address your comments and further improve our work.

---

> > ### Comment · Reviewer_Xkya · 2025-11-22
> > **Response to Authors' Rebuttal**
> >
> > Thanks for the authors' detailed rebuttal. The authors' rebuttal addressed my questions, and I will maintain my positive rating.

---

> > > ### Author Response · Authors · 2025-11-24
> > >
> > > Thank you sincerely for your support! We will remain committed to providing proactive responses to any further questions or inquiries.

---

### Official Review · Reviewer_uV9W · 2025-10-31

**Soundness:** 3
**Presentation:** 3
**Contribution:** 2
**Rating:** 4
**Confidence:** 3

**Summary:**

This paper presents a novel approach to the reranking stage in multi-stage recommender systems. The authors reframe the reranking task as a noise reduction problem, where the goal of the reranker is to "denoise" the scores provided by an earlier, less sophisticated retriever stage. To address this, they propose the Denoising Neural Reranker (DNR), an adversarial framework composed of a denoising reranker and a noise generation module. The framework is optimized through three objectives: an augmented denoising loss that aligns the reranker with user feedback using both observed and synthetic retriever scores; an adversarial loss that encourages the noise generator to produce challenging samples to improve reranker robustness; and a score distribution regularization term to ensure the synthetic scores resemble real retriever scores.

**Strengths:**

1. The conceptualization of reranking as a denoising process is an interesting and creative perspective on the interplay between stages in a cascaded recommendation pipeline. This approach moves beyond treating the stages as independent modules and instead explicitly models their relationship, which is a valuable direction for research in this area.
2. The authors have conducted a thorough set of offline experiments. The ablation studies, which systematically evaluate the contribution of each loss component and the choice of noise generator, provide clear insights into the model's mechanics.

**Weaknesses:**

1. The proposed DNR method is conditioned on the scores from an upstream retriever. This creates a significant risk of inheriting and amplifying any systemic biases (e.g., popularity bias) present in the retriever's output, which could harm the diversity and fairness of the final recommendations. Were any beyond-accuracy metrics evaluated in the offline experiments?
2. The paper's motivation is to improve large-scale, multi-stage industrial recommender systems. However, a key dataset used for evaluation, ML-1M, is small enough that a complex retrieval-reranking pipeline may not be necessary. Could the authors comment on the justification for this architecture on this scale and discuss how they expect the findings to generalize to truly large-scale corpora (e.g., with millions of items).
3. The pattern of increased short-term views (realshow) alongside decreased long-form engagement (watch-time) is a potential red flag. It may indicate that the model is promoting more "clickbait" or superficially appealing content that fails to retain user attention, which could be detrimental to long-term user satisfaction and retention. Presenting this outcome as a clear victory is a misleading interpretation of the complex dynamics of online recommender systems.

**Questions:**

Please refer to weaknesses part.

---

> ### Author Response · Authors · 2025-11-21
> **Response to Reviewer uV9W --- Part 1/2**
>
> We sincerely appreciate your thorough review and thoughtful suggestions. Your concerns are valuable and help us strengthen the paper significantly. Below, we provide detailed responses to address your remaining concerns.
>
> ***1) Bias Amplification and Diversity/Fairness Concerns:***
>
> Thank you for raising this important consideration. We respectfully clarify that DNR is, by its very design, not intended to amplify the retriever scores' bias but only to reduce the scores' "noise" in terms of the user feedback prediction.
>
> Theoretically, biases (including popularity bias) in the retriever’s scores may not align with ground-truth user preferences. This means such biases will be "denoised" if they constitute noise in the score signal. Unlike alternative methods that leverage retriever scores directly, **DNR’s primary advantage lies in denoising unreliable scores, enabled by its noise-aware training paradigm**, as evidenced by Table 2. For example, a specific user may not like a certain video even if it is popular. When the noise-aware DNR finds this noise and corrects it, the resulting prediction becomes more accurate, more personalized, and more diverse.
>
> To better address this concern, we **conduct additional experiments on the diversity of the final recommendations** by comparing the Coverage@1 and coverage@6 metrics of the rerankers' outputs. As shown in Table R3, the experimental results demonstrate that DNR’s denoising capability enables it to mitigate incorrect biases in retriever scores, thereby achieving greater diversity compared to baseline rerankers (e.g., PRM and PIER).
>
> **Table R3:** The diversity performance comparison of DNR and different rerankers.
> | Methods | ML-1M|  | Kuaivideo  |   | Book   |  |
> | :---:   | :-----: | :------: | :------: | :------: | :------: | :------: |
> |        | Coverage@1| Coverage@6 | Coverage@1| Coverage@6 | Coverage@1| Coverage@6|
> | PRM |0.5366| 0.9080|0.7609 |0.9648|0.4364 |0.8584|
> | Pier |0.5429|0.9073|0.7554| 0.9571|0.4238|0.8663|
> | DNR |0.5626 |0.9109|0.7923|0.9702|0.4786|0.8721|
>
>
> Additionally, we also analyzed the diversity metric (i.e., category-based coverage) in the online experiment, which shows 0.00% [-0.03%,+0.03%] change in the A/B test and minimal influence on diversity. This relatively disentangled performance may be related to the post-processing control module that separately controls the recommendation diversity in the online system.
>
> In general, the noise-aware DNR should align with the user feedback, reducing the bias if this bias disagrees with the correct answer. We will include an additional discussion about diversity in Appendix E.
>
> ***2) Concerns about ML-1M and scaling to larger datasets:***
>
> We do have the same concern about ML-1M due to its relatively smaller scale, and that is also why we **included more datasets (Kuaivideo and Book)** and **the online experiment** (in Appendix E.6) to better investigate how DNR behaves in general. Note that the online experiments verify that DNR still works when scaling to 100+ million users and the same level of items.
>
> In ML-1M, we do notice that the Gaussian heuristic is more effective than the Beta heuristic, which is different from all other environments. While we are skeptical about whether this could be attributed to the dataset size, this exceptional behavior reveals that the Beta distribution is empirically a better choice for most recommendation scenarios, but not all.

---

> ### Author Response · Authors · 2025-11-21
> **Response to Reviewer uV9W --- Part 2/2**
>
> ***3) Concerns about decreasing metric and long-term retention:***
>
> **Clarification about metrics:** In our system, all of the following metrics are considered as "short-term" engagement metrics that contribute to the core metric app-time: watch-time, share, like, comment, save, and profile clicks. Typically, the watch-time is not necessarily a long-term metric but rather an engagement metric, since it is directly connected to the user's watching behavior on each video. In contrast, the realshow metric estimates the amount/support of the "engagement", which is usually independently optimized. The "app-time" metric describes how much users have spent on the platform, and it is influenced by all other metrics. Yet, it is not necessarily a bad sign if it slightly decreases, since users' time may have been put to better use by presenting more desired recommendations within a shorter time range.
>
> **Statistical significance:**
> As detailed in Appendix E.6 (Table 11), the slight decline you noted in watch-time (-0.091%), app-time (-0.011%), and share-rate (-0.758%) **was not statistically significant (p > 0.05)**. Empirically, the watch-time has an empirically significant region beyond [-0.16%,+0.16%], the app-time's region beyond [-0.09%, +0.09%], and all other engagement metrics have empirical regions beyond [-1.3%,+1.3%]. In contrast, **our primary optimization was conducted on the binary PCTR (corresponding to realshow metric)**, and achieved a +1.089% gain that was statistically significant (empirically significant region is beyond [-0.5%,+0.5%]), demonstrating the effectiveness of DNR. As we have described in Appendix F.2, achieving performance gains across all online metrics is inherently a **multi-objective task**, which remains a key area for future work. We believe that **applying DNR to other xTR signals** also holds potential for further improvements, and the current result sufficiently supports the effectiveness of DNR.
>
> **Long-term performance:**
> We additionally observed that **the long-term DAU slightly increased by 0.01% [-0.012%,+0.012%]**, which is nearly reaching a statistically significant improvement. This potentially suggests that the slight reduction in viewing duration is likely attributable to users exploring a broader range of clicked videos, while more significant long-term indicators remain positive. In other words, the users may have found their time more effectively used in the system since more valuable information is consumed in a shorter time range. This may also address the concern about "clickbait" videos. We will supplement the long-term DAU in Table 11 of Appendix E.6.
>
> **Clickbait concerns:**
> To further address the concern about the click-bait issue, we additionally observed the "hate-rate" metric, since an increase in click-bait videos would empirically cause an increase in this metric. In our online experiment, we find that this **hate-rate is actually decreased by -0.178% [-0.19%,+0.19%]** in the A/B test (close to statistical significance), which indicates that DNR may help reduce such unwanted videos and improve the experiences of users. This may also contribute to the slightly increased long-term DAU.
>
>
> We sincerely hope that our additional response could address your concerns. If there are any remaining concerns, please let us know, and we will continue to actively address your comments and further improve our work.

---

> ### Author Response · Authors · 2025-11-26
> **Kindly Reminder**
>
> Dear Reviewer uV9W,
>
> We deeply appreciate your valuable feedback and the time you've taken to review our work, especially during this busy period.
>
> We are reaching out to kindly inquire about the current status of your review regarding our submission. We sincerely hope that our responses have adequately addressed your concerns. Furthermore, we are eager to address any additional queries you might have, which will enable us to enhance our work further.
>
> Once again, thank you for your guidance and support.
>
> Best, Authors

---

### Official Review · Reviewer_XnLA · 2025-10-31

**Soundness:** 3
**Presentation:** 3
**Contribution:** 3
**Rating:** 6
**Confidence:** 4

**Summary:**

This paper proposes DNR (Denoising Neural Reranker), an adversarial framework for retriever-aware reranking in multi-stage recommender systems. Unlike prior work focusing on retriever optimization, DNR treats reranking as a noise reduction problem over retriever scores.
It jointly optimizes three objectives — denoising, adversarial noise generation, and score distribution regularization — to align with user feedback. Experiments on public datasets and an industrial system show consistent improvements over strong baselines.

**Strengths:**

• Novel perspective: The paper reformulates reranking as a noise reduction process over retriever scores — an underexplored yet conceptually important viewpoint.
• Comprehensive framework: The proposed approach integrates denoising, adversarial noise generation, and distribution alignment within a unified learning objective, forming a coherent and well-motivated framework.
• Strong empirical results: The method demonstrates consistent and notable improvements across three public benchmarks as well as in a large-scale industrial deployment, supporting its practical effectiveness.

**Weaknesses:**

• Limited retriever diversity: The retriever stage relies solely on a basic Matrix Factorization model. Stronger or more modern retrievers (e.g., dual-tower or ANN-based models) are not evaluated, which limits the generalizability of the conclusions.
• Marginal online gains: The online A/B test reports only a +1.09% improvement in realshow, while other engagement metrics (e.g., share rate, watch time) show slight declines. This raises concerns about the robustness and stability of DNR in real-world settings.

**Questions:**

1. How sensitive is DNR to the quality of the underlying retriever? Would the advantage of denoising diminish when paired with a stronger retriever (e.g., ANN-based)?
2. In the online experiment, some engagement metrics decrease slightly — could this reflect overfitting to short-term user feedback signals?
3. Could the denoising concept be extended to a joint retriever–reranker optimization framework, where both components are trained end-to-end?

---

> ### Author Response · Authors · 2025-11-21
> **Response to Reviewer XnLA --- Part 1/2**
>
> We are extremely grateful for your positive and insightful review. We have detailed our responses point-by-point to your comments. We sincerely hope that our additional response has adequately addressed your concerns.
>
> 1) ***Limited Retriever Diversity:***
>
> To better illustrate the generalizability of DNR across different retrievers, we implemented the dual-tower (DT) model as a stronger retriever. The comparison results in the retrieval stage are summarized in Table R1, and the comparison results in the reranking stage are summarized in Table R2. We can see that DT+DNR outperforms DT+PRM, validating the generalizability of our method and the sustained advantage of the denoising framework.
>
> **Table R1:**  The AUC performance comparison of **different retrievers**.
> | Methods | ML-1M | Kuaivideo | Book |
> | :---: | :---: | :---: | :---: |
> | MF | 79.20| 89.62 |88.82|
> | DT |79.86|90.38| 90.01|
>
> **Table R2:** The performance comparison of **rerankers** on different retrievers.
> | Methods | ML-1M|  | Kuaivideo  |   | Book   |  |
> | :---:   | :-----: | :------: | :------: | :------: | :------: | :------: |
> |        | NDCG@6| MAP@6 | NDCG@6| MAP@6 | NDCG@6| MAP@6    |
> | MF+PRM | 72.85 | 62.21|55.93 |42.97|76.88 |66.44 |
> | MF+PIER | 75.99 | 65.98|65.11 |52.55|80.22| 71.62 |
> | MF+DNR | 77.67 | 68.00|70.15|58.12|83.53|75.54|
> | DT+PRM | 76.68| 67.37 |65.77 |53.41| 81.18 | 72.72 |
> | DT+PIER | 76.36 | 66.35| 68.34 | 56.26 | 81.55 | 72.79 |
> | DT+DNR | 77.96| 68.78 |70.96|58.87| 84.42 | 77.11 |
>
> 2) ***Diminishing performance with advanced retrievers:***
>
> Theoretically, there will always be an empirical error that upper bounds the overall recommender system, which aims to minimize the "optimizable error", and there will be no room for the reranker to "denoise the scores" if the retriever itself achieves this bound. Yet, this theoretical bound is still impractical for a single retriever in most industrial platforms, due to various reasons including (but not limited to) the computational bottleneck, multi-task trade-off, and filtering on a large-scale dynamic candidate pool. Typically, it may contradict our assumption that the retriever uses a simpler model design than the reranker for the latency requirement. On the other hand, a more advanced retriever means a smaller remaining error that can be optimized, which indicates that the reranker will solve a simpler denoising problem and a simpler reranker design. In this case, a diminishing marginal gain only indicates a diminishing computational requirement in the reranking stage. In general, we believe that the denoising framework holds, and how to achieve an optimal architectural balance between different stages may need further investigation.

---

> ### Author Response · Authors · 2025-11-21
> **Response to Reviewer XnLA --- Part 2/2**
>
> 3) ***Marginal online gains:***
>
> - **Clarification (Statistical Significance):** As detailed in Appendix E.6 (Table 11), the slight declines you noted in watch-time (-0.091%) and share-rate (-0.758%) were not statistically significant (p > 0.05). Empirically, the watch-time has an empirically significant region beyond [-0.16%,+0.16%], and all other "interaction" metrics have an empirically significant region beyond [-1.3%,+1.3%]. Here, "interaction" metrics include watch-time, share, like, comment, save, and profile clicks, and they all contribute to the core metric "app-time" (statistically significant region beyond [-0.09%,+0.09%]). In contrast, the realshow metric determines the support of the "interaction" and the "app-time", which is usually independently optimized.
> - **Primary Goal Achieved:** Our primary optimization was **conducted on the binary PCTR (corresponding to realshow)** metric, achieving a +1.089% gain that was statistically significant (empirically significant region is beyond [-0.5%,+0.5%]), demonstrating the effectiveness of DNR. As we have described in Appendix F.2, achieving performance gains across all online metrics is inherently a **multi-objective task**, which remains a key area for future work. We believe that **applying DNR to other xTR signals** also holds potential for further improvements.
> - **Short-term vs. long-term engagement:** While the watch time exhibited a slight decrease, we additionally observed that **the long-term DAU slightly increased by 0.01% [-0.012%,+0.012%]**, which is nearly reaching a statistically significant improvement. This potentially suggests that the slight reduction in viewing duration is likely attributable to users exploring a broader range of clicked videos, while more significant long-term indicators remain positive. In other words, the users may have found their time more effectively used in the system since more valuable information is consumed in a shorter time range. We will supplement the long-term DAU in Table 11 of Appendix E.6.
> Conclusion: We significantly improved our main objective metric without causing any statistically significant harm to other long-term metrics, indicating a "relatively independent" optimization scheme. This is widely considered a successful and positive outcome in a real-world industrial setting. We will meticulously revise Appendix E.6 to enhance the clarity.
>
> 4) ***Joint Retriever-Reranker Optimization:***
>
> In this work, we merely stated that "retriever-aware reranker" could be a complementary piece to the "reranker-aware retriever" work, but we agree that optimizing the two components end-to-end jointly is a very promising direction for future research. The potential lies in tackling the core multi-stage challenges: effectively aligning the divergent training objectives, reliably back-propagating non-uniform feedback signals across the stages, and allowing a flexible model switch for any stage. We will extend the "multi-stage vs. end-to-end" section in Appendix F.2 to further discuss this possibility.
>
>
> We sincerely hope that our additional response could address your concerns. If there are any remaining concerns, please let us know, and we will continue to actively address your comments and further improve our work.

---

> ### Author Response · Authors · 2025-11-26
> **Kindly reminder**
>
> Dear Reviewer XnLA,
>
> We deeply appreciate your valuable feedback and the time you've taken to review our work, especially during this busy period.
>
> We are reaching out to kindly inquire about the current status of your review regarding our submission. We sincerely hope that our responses have adequately addressed your concerns. Furthermore, we are eager to address any additional queries you might have, which will enable us to enhance our work further.
>
> Once again, thank you for your guidance and support.
>
> Best, Authors

---

### Author Response · Authors · 2025-11-28
**General Response**

Dear ACs and Reviewers,

We sincerely thank all ACs and reviewers for the time and constructive feedback. We would like to summarize **our contribution**, the **strengths** acknowledged by reviewers, the reviewers' **main concerns**, and **the responses** we have made to address all the concerns.

**Contributions:**
- **Theoretical Findings:** We formally define reranking as a noise reduction problem (linking informative-yet-noisy retriever scores to user feedback alignment) and prove that naive retriever score using strategies optimize a suboptimal likelihood upper bound. We then mathematically derive DNR’s three-objective framework via variational inference—addressing the underexplored retriever-aware reranker design gap.
- **New Method:** We propose DNR, a novel retriever-aware reranking framework with noise generator, decomposing user feedback alignment into three objectives: denoising (observed/synthetic) retriever scores, adversarial learning for synthesis of hard samples, and aligning synthetic-real retriever score distributions.
- **Comprehensive validation:** Extensive offline experiments and online a/b test show DNR outperforms state-of-the-art baselines and boosts industrial "realshow" by 1.089% (statistically significant).

**Strengths:**
- **Novelty and Motivation**: A novel perspective reformulating reranking as a noise reduction process (Reviewer `XnLA`, `uV9W`, `6Pyu`), which is empirically and theoretically well-motivated (Reviewer `XnLA`, `Xkya`).
- **Quality and Soundness**: Comprehensive framework integrating denoising, adversarial generation, and distribution alignment (Reviewer `XnLA`, `6Pyu`), with clear writing and derivation (Reviewer `Xkya`).
- **Experimental Validation**: Extensive offline experiments and a large-scale industrial deployment (Reviewer `XnLA`, `Xkya`, `uV9W`, `6Pyu`), demonstrating consistent improvements over baselines.

**Main concerns raised by reviewers:**
- **Ablation on retrievers**: Performance with stronger/more diverse retrievers (Reviewer `XnLA`)
- **More recent baselines**:comparison with "Comprehensive List Generation" (Reviewer `6Pyu`).
- **Diversity and bias**: Concerns regarding potential bias amplification from the retriever and performance on diversity metrics (Reviewer `uV9W`).
- **Online gains**: Discussion on the metric gains (e.g., RealShow vs. Watch-time) in the online A/B test (Reviewer `XnLA`, `uV9W`, `6Pyu`).
- **Presentation details**: Clarity of the motivation in figures (Reviewer `Xkya`) and long sentences (Reviewer `6Pyu`) .The justification for distribution choices and the neglect of the $\delta_x$ term in the loss decomposition (Reviewer `6Pyu`)

**Our responses to reviewers' concerns:**
- **Ablation on retrievers**: We conducted additional experiments using a Dual-Tower (DT) model as a stronger retriever (as mentioned by Reviewer `XnLA`), where DT+DNR continued to outperform baselines like DT+PRM.
- **More recent baselines**: We also compared DNR against the requested baseline MG-E (CLIG), demonstrating that DNR achieves state-of-the-art performance.
- **Diversity and bias**: We clarified that the noise-aware DNR denoises incorrect biases rather than amplifying them. We also added a diversity analysis using Coverage@1 and Coverage@6 metrics, showing that DNR **achieves greater diversity compared to baselines**. In the online experiment, the diversity metric has no change, further proving DNR does not necessarily amplify the bias or reduce diversity.
- **Online gains**: 1) We clarified the details of the **statistical significance of gains** on different metrics. 2) In addition to the optimized metric "realshow", we also noted **a slight increase in "long-term DAU" (long-term metric), and a drop in "hate-rate" (clickbait indicator)**, confirming our success. 3) Additionally, we remind that balancing gains across all online metrics is inherently **a multi-objective task for future work**.
- **Presentation details**: We revised Figure 1 to incorporate a noise distribution analysis that explicitly visualizes the denoising effect, and refined several long sentences to improve readability. And, we provide more detailed justification for the use of the Beta distribution and the model-based noise generator. We also provided the theoretical proof for the neglect of term $\delta_x$ .

**Reviewers' feedback:** We are grateful to Reviewer `Xkya` for confirming that their questions were addressed and maintaining a positive rating (Score: 8) in 22 Nov. We believe the detailed supplementary experiments and clarifications presented in our rebuttal have satisfactorily addressed the concerns raised by Reviewers `XnLA`, `uV9W`, and `6Pyu`—including those pertaining to baselines, retriever diversity, and online metrics.

We have incorporated all additional experiments and analyses into the revised paper. We hope the improvements made will be taken into consideration. We sincerely appreciate your valuable time!

Best regards,

Authors of paper #3873

---

### Public Comment · ~Zixuan_Tang1 · 2026-05-22
**Request for ML-1M preprocessing script / preprocessed data files**

Dear authors,

Thank you for releasing the code of Robust Denoising Neural Reranker for Recommender Systems — the framework is very inspiring.

I am trying to reproduce the ML-1M results reported in the paper, but I noticed that the repository only contains the training scripts (stage1_main.py, stage2_main_our.py, etc.) and assumes the existence of the following files under dataset/ml-1m/:

train_stage1_new.df

test_stage1.df

train_processed.df

test_processed.df

data_statis.df

popularity_score.pt

However, neither the preprocessing script nor the preprocessed .df / .pt files are included in the repo, and the README does not describe how the raw MovieLens-1M ratings are converted into these files (e.g. history length, slate construction strategy, negative sampling protocol for test_stage1.df, train/test split, label binarization threshold, etc.).

Could you kindly:

Release the preprocessing script you used for ML-1M (and ideally for amazon_book / kuaireal as well), or

Share the preprocessed dataset/ml-1m/ folder directly (e.g. via Google Drive Hugging Face Zenodo)?

This would greatly help the community faithfully reproduce the numbers reported in Table X of the paper. Thanks a lot for your time!

Best regards

---

### Meta-Review · Area_Chair_geuD · 2026-01-06

**Summary:**

The reviewers generally praised the novel perspective of reframing reranking as a noise reduction problem. However, several critical concerns informed the final evaluation:
1. Limited Evaluation Diversity: Reviewer noted that the initial experiments relied on a simple Matrix Factorization retriever, questioning if the benefits would hold with modern retrievers like Dual-Tower models
2. Marginal Online Gains and "Clickbait" Concerns: Multiple reviewers expressed concern over the online A/B test results (only marginal improvement) . Reviewer uV9W specifically worried this indicated a shift toward "clickbait" content that fails to retain long-term user interest.
3. Missing & small-scale Baselines: Reviewer identified a missing recent baseline (MG-E/CLIG) and requested better justification for the choice of the Beta distribution. Also the ML-1M dataset seems to be too small.

**Reviewer Concerns:**

The authors provided a very comprehensive rebuttal that successfully addressed the majority of technical and empirical gaps:
- Retriever Generalizability: The authors added experiments using a Dual-Tower (DT) model. Results showed that DT+DNR consistently outperformed DT+PRM, proving the method's effectiveness even with stronger retrievers
- Missing Baselines: The authors included a comparison with the MG-E (CLIG) baseline. DNR outperformed this recent method across all three datasets
- Clickbait Evidence: The authors introduced new online metrics: a slight increase in long-term DAU and a decrease in "hate-rate" (a proxy for clickbait), effectively countering the "short-term clickbait" argument

Outstanding Concerns
- Metric Trade-offs: While the authors proved the declines in engagement metrics (watch-time, etc.) were not statistically significant, the fact that the gains were heavily concentrated in one metric ("realshow") remains a point of discussion for industrial applicability
- Small-Scale Dataset Limitations: The concern regarding the necessity of a complex multi-stage pipeline for a dataset as small as ML-1M remains a conceptual critique, though partially mitigated by the inclusion of larger datasets and online tests.

**Reviewer Scores:**

Reviewer Xkya (Initial 8): No change.
Reviewer XnLA (Initial 6): Potential +1
Reviewer uV9W (Initial 4): potential +1
Reviewer 6Pyu (Initial 4): potential +1

---

### Decision · Program_Chairs · 2026-01-26

Accept (Poster)